# Association of adverse prenatal exposure burden with child psychopathology in the Adolescent Brain Cognitive Development (ABCD) Study

**Joshua L. Roffman**[1]\*, **Eren D. Sipahi**[1], **Kevin F. Dowling**[1], **Dylan E. Hughes**[1], **Casey E. Hopkinson**[1], **Hang Lee**[2], **Hamdi Eryilmaz**[1], **Lee S. Cohen**[1], **Jodi Gilman**[1], **Alysa E. Doyle**[1,3], **Erin C. Dunn**[1,3]

**1** Department of Psychiatry, Massachusetts General Hospital and Harvard Medical School, Boston, MA, United States of America, **2** Massachusetts General Hospital Biostatistics Center, Massachusetts General Hospital, Boston, MA, United States of America, **3** Center for Genomic Medicine, Massachusetts General Hospital, Boston, MA, United States of America

\* jroffman@partners.org

**Data Availability Statement:** Data can be accessed through the NIMH Data Archive, https://nda.nih.gov/abcd.

## Abstract

### Objective

Numerous adverse prenatal exposures have been individually associated with risk for psychiatric illness in the offspring. However, such exposures frequently co-occur, raising questions about their cumulative impact. We evaluated effects of cumulative adverse prenatal exposure burden on psychopathology risk in school-aged children.

### Methods

Using baseline surveys from the U.S.-based Adolescent Brain and Cognitive Development (ABCD) Study (7,898 non-adopted, unrelated children from 21 sites, age 9–10, and their primary caregivers), we examined 8 retrospectively-reported adverse prenatal exposures in relation to caregiver-reported total and subscale Child Behavior Checklist (CBCL) scores. We also assessed cumulative effects of these factors on CBCL total as a continuous measure, as well as on odds of clinically significant psychopathology (CBCL total $\geq$60), in both the initial set and a separate ABCD sample comprising an additional 696 sibling pairs. Analyses were conducted before and after adjustment for 14 demographic and environmental covariates.

### Results

In minimally and fully adjusted models, 6 exposures (unplanned pregnancy; maternal alcohol, marijuana, and tobacco use early in pregnancy; pregnancy complications; and birth complications) independently associated with significant but small increases in CBCL total score. Among these 6, none increased the odds of crossing the threshold for clinically significant symptoms by itself. However, odds of exceeding this threshold became significant with 2 exposures (OR = 1.86, 95% CI 1.47–2.36), and increased linearly with each level of

**Funding:** JR: National Institutes of Health (R01MH101425, R01MH124694) David Judah Fund Rappaport Foundation MGH Department of Psychiatry MGH Executive Committee on Research MGH Early Brain Development Initiative ED: National Institutes of Health (R01MH113930) LC: Ammon-Pinizzoto Center for Women's Health at MGH The funders had no role in study design, data collection and analysis, decision to publish, or preparation of the manuscript.

**Competing interests:** The authors have declared that no competing interests exist.

exposure (OR = 1.39, 95% CI 1.31–1.47), up to 3.53-fold for ≥4 exposures versus none. Similar effects were observed in confirmatory analysis among siblings. Within sibling pairs, greater discordance for exposure load associated with greater CBCL total differences, suggesting that results were not confounded by unmeasured family-level effects.

## Conclusion

Children exposed to multiple common, adverse prenatal events showed dose-dependent increases in broad, clinically significant psychopathology at age 9–10. Fully prospective studies are needed to confirm and elaborate upon this pattern.

## Introduction

Long held neurodevelopmental theories posit that risk for serious mental illness begins early in life. In particular, the fetal environment is thought to lay a foundation for such risk, in concert with genetic loading [1, 2]. This insight has given rise to Barker's Developmental Origins of Disease model and others [3, 4], which describe a prenatal "programming" of risk for medical and psychiatric disorders that emerge even well after birth.

Birth cohort studies enable researchers to test associations between prenatal exposures and a range of neurodevelopmental and psychiatric outcomes [5]. For example, some well-known birth cohort studies have shown that offspring exposed to starvation *in utero* during the Dutch Hunger Winter [6] and the Chinese famine of 1959–61 [7] had a 2-fold increased risk of psychotic illness two decades later. Across both retrospective and prospective studies, adverse prenatal exposures that occur more frequently, such as pregnancy or birth complications [8, 9], prematurity [10], maternal infections (including both serious infections such as influenza, and more minor ones such as urinary tract infections) [11, 12], and maternal substance or tobacco use [13–20] have also associated with a range of psychopathology, including disorders that emerge during childhood. However, these more prevalent exposures tend to have more modest effects (typically >50% increase in odds for psychopathology). For example, mildly reduced body mass index during pregnancy has been found to associate with a 21% increased risk of nonaffective psychosis, which is a 10-fold lower magnitude of effect compared to that observed for famine [21]. Yet, even for these more common events, dosing of exposure seems to matter. Indeed, heavy alcohol use throughout pregnancy has been found to associate with greater psychopathology than occasional use [22].

Whether exposure to multiple common insults during pregnancy also exerts a dose-dependent risk for psychopathology remains unclear. There are few models that account for effects of multiple risk factors on such risk, although polygenic risk scoring (PRS) is perhaps the most notable. PRS studies demonstrate cumulative, linear effects across thousands of genetic variants of small effect; however, additive effects of even the strongest (e.g., genome-wide significant) common genetic variants are modest [23]. In contrast, given the larger effect sizes attributed to individual prenatal environmental exposures, it is possible that only a small number of such exposures occurring in linear combination could substantially increase risk–not only for increased dimensional symptoms, but potentially for crossing the threshold into clinically significant psychopathology.

While cumulative effects of adverse *postnatal* exposures have been observed in children [24, 25] and adults [26], the loading burden of experiencing multiple types of adverse prenatal events has been studied only in a relatively small sample of preschool-aged children [27] and children with autism [28]. Further, adverse prenatal exposures frequently co-occur, reflecting

both causal links (e.g., maternal substance use resulting in other pregnancy complications [29]), and shared background factors (e.g., poverty increasing risk for both maternal substance use and other pregnancy complications [30]). Similarly, just as diverse categories of psychopathology share substantial genetic loading [31], the clinical effects of prenatal environmental insults may be non-specific [32–34]. Efforts to account for overlapping risk factors are needed to decipher the individual and joint role of prenatal exposures on psychopathology risk and develop intervention targets.

Disentangling the effect of these prenatal exposures on psychopathology symptoms is especially challenging in studies of children, who experience rapid neurodevelopmental changes. The Adolescent Brain and Cognitive Development (ABCD) Study [35], which has enrolled 11,875 9- and 10-year-old children across 21 U.S.-based sites, provides the opportunity to study a cohort that is developmentally narrow, but also broadly and systematically characterized in regard to psychopathology and related environmental risk factors.

We leveraged baseline data from the ABCD Study to relate cumulative burden or loading of adverse prenatal exposures, obtained through retrospective report, to dimensional measures of psychopathology. The use of dimensional measures enabled assessment of early and subdiagnostic psychopathology in school-aged children, and leveraged continuous variance in these traits across the population. The ABCD Study provides extensive phenotyping of both prenatal and postnatal exposures that have associated with psychopathology risk (e.g., trauma, family conflict) in prior studies [36–39], as well as of dozens of demographic and environmental features that potentially confound these relationships. The ABCD Study also includes siblings, and analysis of sibling pairs enables additional control over potential confounding effects of unmeasured family-level variables. If the association is causal, siblings who are discordant for exposures should show greater differences in psychopathology. Conversely, if the discordant siblings do not differ in psychopathology, the observed association is explained by unmeasured genetic and environmental factors. For example, the results of studies with sibling design have supported causal relationships between prenatal exposure to tobacco [40], alcohol [41], and obstetrical complications [42] and subsequent psychopathology.

We hypothesized that [1] multiple individual adverse prenatal exposures would associate independently with increased dimensional psychopathology; and [2] increased additive loading for such individual adverse prenatal exposures would associate with greater odds of psychopathology, as measured both dimensionally and via thresholded indices of clinically-relevant psychopathology [43]. We treated participants who did not have siblings as an initial sample, and then separately examined sibling pairs for the purposes of validation and control for family-level confounders.

## Method

### Participants

We analyzed baseline data from a total of 9,290 participants in the ABCD study, a longitudinal cohort of 11,875 children from 21 research sites across the United States. All data were obtained from the NIMH Data Archive, Curated Annual Release 2.0. General inclusion and exclusion for the ABCD study are described elsewhere [44, 45]. In brief, 9- to 10-year-old children were recruited from the community, had no contraindications to MRI scanning, and were excluded if they were not fluent in English; had a history of major neurological disorders, traumatic brain injury, or extreme prematurity; or carried a diagnosis of schizophrenia, moderate to severe autism spectrum disorder, intellectual disability, or substance use disorder.

The current analysis included only non-adopted children from singleton pregnancies (twins were excluded given known differences in pre- and postnatal life compared to

singletons) and children with valid Child Behavior Checklist (CBCL) scores. Children were grouped based on whether they did (n = 1,392) or did not (n = 7,898) have a sibling who also participated in the study (S1 Fig in S1 File). Partitioning children based on their sibling status enabled us to use the non-sibling group as an initial set and the sibling group as a non-overlapping validation set; the latter was leveraged to examine effects of discordant prenatal exposures on CBCL scores, thus controlling for unmeasured family-level confounders. Given that enrollment occurred over a 2 year period, and age at enrollment was constrained to 9.0 to 10.9 years, all sibling pairs differed in age by less than 4 years.

IRB approval for the ABCD study is described in Auchter et al. [46]. Most ABCD research sites cede approval to a central Institutional Review Board (cIRB) at the University of California, San Diego, with the remainder obtaining local IRB approval. All parents provided written informed consent and all children provided assent.

## Measures

All measures were collected at the baseline ABCD in-person study visit from the primary caregiver, who was also the biological mother in 88.1% of Non-sibling and 88.7% of Sibling participants.

**Prenatal exposures.** Primary prenatal exposures of interest were obtained based on caregiver recall and recorded in the ABCD *Developmental History Questionnaire*. The following 15 exposures were extracted for each individual and coded as present or absent, consistent with dichotomous analyses used in previous studies (e.g., [9, 18, 25, 26]):

[1] unplanned pregnancy;

[2–6] maternal use of alcohol, tobacco, marijuana, cocaine, or opiates, *before* pregnancy was recognized ("early" exposure);

[7–11] maternal use of alcohol, tobacco, marijuana, cocaine, or opiates, *after* pregnancy was recognized ("late" exposure);

[12] Caesarian section;

[13] pregnancy complications (coded as present if ≥1 of 13 listed obstetric complications, e.g., gestational diabetes or preeclampsia, occurred, see Methods in S1 File for complete list);

[14] birth complications (coded as present if ≥1 of 8 listed complications, e.g., jaundice, oxygen requirement, occurred, see Methods in S1 File for complete list);

[15] pre-term birth (coded as present if birth occurred before gestational week 37).

Exposures present in less than 5% of the sample were dropped from the analysis, excepting one scenario: if early substance exposure occurred in at least 5% of cases, late substance exposure was also included regardless of frequency. This approach was necessary given the high co-occurrence with main predictors of interest (i.e., early use of these substances), potentially different teratogenic effects, and related confounding potential (see also S3 Table in S1 File). The 5% threshold was chosen a priori to minimize the risk of overfitting in exposure subgroups, given the total number of covariates.

Specifically, opiate or cocaine use during pregnancy (early or late) was reported in less than 5% of the sample, as were late alcohol, tobacco, or marijuana use. This resulted in using 8 adverse prenatal exposures for primary analysis (unplanned pregnancy; early alcohol, tobacco, or marijuana exposure; pregnancy complications; birth complications; preterm birth; and Caesarean section). Late exposure to alcohol, tobacco, or marijuana were treated as covariates of no interest, as above.

**Psychopathology symptoms.** The CBCL [43] is a widely-used tool to assess dimensional psychopathology syndromes and related symptoms. In addition to 8 syndrome scales (Anxious/Depressed, Withdrawn/Depressed, Somatic Complaints, Social Problems, Thought Problems, Attention Problems, Aggressive Behavior, Rule-breaking Behavior) the CBCL contains 2 broad-band scales (Internalizing and Externalizing Problems) and a total score, all recorded as t-scores.

**Covariates.** We included a total of 10 covariates reflecting demographic, socioeconomic, and environmental variables that could potentially covary with primary prenatal exposures of interest and/or CBCL scores. These data were extracted from the ABCD *Demographics Survey*, *Developmental History Questionnaire*, *K-SADS Parent Diagnostic Interview*, and *Family History Assessment* and included:

[1] child's age (months);

[2] child's sex;

[3] child's race (Caucasian/non-Caucasian);

[4] child's ethnicity (Latinx/non-Latinx/unsure);

[5] presence or absence of a partner for the primary caregiver;

[6] caregiver income (averaged across both primary caregivers, if present; see Methods in S1 File);

[7] caregiver education (averaged across both primary caregivers, if present; see Methods in S1 File);

[8] maternal age at birth;

[9] neighborhood safety;

[10] presence or absence of an older sibling.

In addition, study site (n = 21 sites) was also included as a nominal random effect in all models. Random, rather than fixed, effect modeling was used due to variation in sample size among sites, and to provide a statistical basis for generalizing results beyond the study sites [47].

Sensitivity analyses also included as covariates postnatal exposures that have been associated with psychopathology risk [36–39], but that were not considered potential confounders, as they could not directly influence prenatal exposures. These sensitivity analyses included (in addition to the 10 listed above):

[11] total weekday screentime;

[12] total weekend screentime (which is differentially associated with psychopathology, see [39];

[13] Family Conflict Subscale of the Family Environment Scale;

[14] child's exposure to zero vs. ≥1 significant traumas (see Methods in S1 File)

## Statistical analysis

Analyses were conducted using R version 3.6.1. Missing data, assumed to be missing at random (MAR), was imputed using multiple imputation by chained equations (mice package 3.11.0), which imputes individual variables according to their own distribution and requires

an imputation method to be assigned to each variable. Cluster-level effects of site were accounted for by creating separate regression coefficients for each site to be used in the imputation model (i.e. a fixed effects approach). Siblings were imputed in wide format and flipped back to long format for subsequent analyses to account for cluster effects of family. Continuous and ordinal variables were imputed with predictive mean matching and dichotomous variables with logistic regression. All variables that were included in later regressions—with the exception of transformed variables—were selected as predictors for the imputation models. 200 datasets were imputed from which parameter estimates were pooled to derive beta weights, confidence intervals, and p-values, per guidelines described by Rubin [48]. This method accounts for variance both within and across the imputation sets (i.e., additional variance due to missing data). We also performed sensitivity analyses to compare group means/distributions and primary results from the full (imputed) and non-imputed data sets.

**Initial group.** Within the initial (Non-sibling) group, general linear models were used to determine which of the 8 primary adverse prenatal exposures that were reported in at least 5% of the sample (i.e., unplanned pregnancy; early alcohol, tobacco, or marijuana exposure; pregnancy complications; birth complications; preterm birth; and Caesarean section, as above) associated with differences in total CBCL score. Importantly, both here and in subsequent analyses, these factors were entered simultaneously (along with covariates) to determine the independent effects of each exposure.

To better isolate effects of the 8 exposures on CBCL scores from those of other covariates, we conducted both minimally and fully adjusted models. The minimally adjusted model included the 8 exposures of interest, site (as a random effect), and late exposure to alcohol, tobacco, or marijuana as covariates, as described in the Prenatal Exposures section above. The fully adjusted model also included as covariates the 10 demographic, socioeconomic, and other exposure-related factors, listed in the Covariates section above, that potentially confound relationships between psychopathology and prenatal exposures.

Next, to assess effects of the 8 adverse prenatal exposures on odds of <u>clinically relevant</u> psychopathology, CBCL total scores were recoded as being within (<60) versus above the threshold for borderline clinical significance, as previously defined [43, 49]. Generalized linear mixed model (GLMM) via the lme4 package in R provided estimates of the odds of CBCL total ≥60 based on the minimally and fully adjusted models described above. Linear mixed models were used instead of ordinary logistic regression to enable analysis of random effects. Again, all exposures and covariates were included simultaneously in the model, enabling us to identify measures that were independent predictors of elevated CBCL score.

Finally, to determine the cumulative odds of CBCL total ≥60 as a function of loading for adverse prenatal exposures, the GLMM was repeated, substituting the integer sum of a child's prenatal exposures; this loading score was calculated using only exposures that were shown to associate with increased CBCL total scores in the fully adjusted multivariate regression models (above). We used this approach to generate additive exposure loading scores because [1] each of the included individual exposures had been previously associated with adverse neurodevelopmental outcomes in previous studies (as well as in this one); [2] it enabled development of a clinically intuitive and straightforward risk score analysis that aggregated known risk factors while accounting for overlapping variance; and [3] given the sample size there were no concerns about overfitting and therefore no need to reduce the number of dimensions. Minimally and fully adjusted models were evaluated as above. Main analyses set alpha at .05, two-tailed. Post hoc analyses determined the effects of adverse prenatal exposure load on individual and broad band CBCL scales as well as risk for clinically significant psychopathology (score ≥65 for individual syndrome scales and ≥60 for broad-band scales) [43], using the family-wise error rate to control for multiple comparisons. Additional sensitivity analyses also included

post-natal exposures that could influence psychopathology risk (see Covariates section, covariates #11–14 above).

**Validation group.** Analyses within the non-overlapping validation (Sibling) group carried forward the same exposure load approach (i.e., children were assigned a load score based on the number of exposures that were associated with CBCL total scores in the non-Sibling analysis.) GLMM was used to determine the effect of exposure load on the odds of CBCL total $\geq$60, using the same minimally and fully adjusted models as before, with the exception that Family ID was now included as an random effect (variance component covariance type). As in the Initial group, additional sensitivity analyses also included post-natal exposures that could influence psychopathology risk (see Covariates section, covariates #11–14 above).

Within-sibling pair analysis was conducted to control for unmeasured genetic and postnatal confounders, using a multilevel modeling method [50]. Exposure load for each sibling within each family was calculated by subtracting the average load across both siblings from the individual load for each sibling. As such, siblings with identical loads would each have an individual load of zero, while those with discordant loads would have equal and opposite positive (i.e., sibling with higher load) or negative (i.e., sibling with lower load) individual load values, respectively. The effects of individual exposure load on CBCL total score were determined using a linear model, controlling for family-level average exposure load, as well as three other factors that could differ between siblings (sex, age, and maternal age at birth) and family ID. To account for heteroscedasticity introduced by cluster effects of site, standard errors and accordingly, p-values and confidence intervals, were adjusted using the Huber-White robust sandwich estimator.

Results reflect fully adjusted models unless otherwise noted. For comparison, minimally adjusted models are also presented within selected Tables.

## Results

Characteristics of the Non-sibling and Sibling groups are provided in Table 1. Group averages were consistent between imputed and non-imputed data (S1 and S2 Tables in S1 File).

### Initial (non-sibling) group

Adverse prenatal exposures were highly correlated with each other, and with demographic, socioeconomic, and postnatal exposures (S3 Table in S1 File). Of the 8 primary prenatal

**Table 1. Characteristics of non-sibling and sibling groups.**

| CONTINUOUS FACTORS | Non-sibling group (N = 7,898) | | Sibling group (N = 1,392) | |
|---|---|---|---|---|
| | Mean | SD | Mean | SD |
| Age (months) | 118.5 | 7.3 | 118.4 | 8.8 |
| CBCL total t-score | 46.4 | 11.3 | 45.1 | 11.3 |
| DICHOTOMOUS FACTORS | N (%) | | N (%) | |
| Sex (female) | 3,708 (46.9) | | 688 (49.4) | |
| High CBCL ($\geq$60) | 1,022 (12.9) | | 150 (10.8) | |
| Unplanned pregnancy | 3157 (40.0) | | 577 (41.5) | |
| Early alcohol exposure | 2,177 (27.6) | | 276 (19.8) | |
| Early tobacco exposure | 1,112(14.1) | | 158 (11.3) | |
| Early marijuana exposure | 513 (6.5) | | 66 (4.7) | |
| Complicated pregnancy | 3,238 (41.0) | | 524 (37.6) | |
| Complicated birth | 1,952 (24.7) | | 333 (23.9) | |
| Preterm birth | 637 (8.1) | | 100 (7.1) | |
| Caesarian section | 2,517 (31.9) | | 373 (26.8) | |

exposures studied, 6—unplanned pregnancy, early maternal alcohol, tobacco, or marijuana use, pregnancy complications, birth complications—associated independently with increased CBCL total score in both minimally and fully adjusted models (p's ≤.017; S4 and S5 Tables in S1 File). Results were similar in magnitude and statistical significance in a sensitivity analysis restricted to the 6,271 individuals with complete data (S6 Table in S1 File). A total of 1,022 (12.9%) individuals had CBCL total scores above normal (≥60). Four of these 6 exposures (all but early alcohol and marijuana use) independently associated with increased odds of CBCL total ≥60, with odds ratios ranging from 1.35 to 1.63 (p's < .001, S7 Table in S1 File).

These 6 factors were summed to generate a cumulative exposure load (ranging from 0 to 6) for each individual. In fully adjusted models there was a significant linear effect of exposure load on risk for both increased CBCL as a continuous measure (1.94 CBCL points per exposure, 95% CI 1.72 to 2.16, p < .001, S8 Table in S1 File) and risk for CBCL total ≥60 (OR 1.39 per exposure, 95% CI 1.31 to 1.47, p < .001, Table 2). Specifically, compared to children with no adverse prenatal exposures, those with ≥2 exposures had significantly greater odds of CBCL total ≥60 (p's ≤.001), and children with 4 or more factors had a 3.53 (95% CI 2.62 to 4.76) greater odds of CBCL ≥60 compared to those with no exposures. In absolute terms, 29% of children exposed to 4 or more factors exhibited CBCL total ≥60, compared to 7% of children with no exposures (Fig 1A). Similar exposure load effects were observed across all individual syndrome and broad-band CBCL scales (p's ≤.001; Fig 2, S9 Table in S1 File).

## Validation (sibling) group

Total load for the same 6 exposures was calculated for each child in the sibling group. Controlling for the same covariates as in the non-sibling analysis, and introducing family ID as an additional random effect, there were again significant linear effect of prenatal adversity load on CBCL total score as a continuous measure (1.86 CBCL points per exposure, 95% CI 1.31 to 2.42, p < .001, S10 Table in S1 File) and risk for CBCL total ≥60 (OR 1.61 per exposure, 95% CI 1.30 to 2.01, p < .001, Table 3; see also Fig 1B). As in the Non-sibling group, compared to those with a load of zero, children with ≥2 exposures had significantly higher odds of CBCL total ≥60 (p's ≤.004, Table 3). There was an approximate 7-fold increased odds of CBCL total ≥60 for those children with ≥4 exposures compared to those with no exposures (OR = 6.88, 95% CI 2.26 to 20.94, p = 0.001). Further, we repeated the analysis without including family ID as a random effect to determine the extent that unmeasured family confounding influenced the results. Resultant odds ratios and confidence intervals were similar (S11, S12 Tables in S1 File) suggesting that effects of family-level confounders were minimal.

In an additional sensitivity analysis, we added 4 additional covariates representing postnatal exposures that can influence psychopathology. Results were consistent with, but slightly weaker than in the fully adjusted model (S13, S14 Tables in S1 File).

**Table 2. Effect of adverse prenatal exposure load on odds of CBCL total score ≥60 in the initial (non-sibling) sample.**

| EXPOSURE LOAD, N | Odds of CBCL total ≥60 (minimally adjusted) | | Odds of CBCL total ≥60 (fully adjusted) | |
|---|---|---|---|---|
| | Odds ratio (95% CI) | p | Odds ratio (95% CI) | p |
| 0, N = 1,640 | Reference | – | Reference | – |
| 1, N = 2,712 | 1.26 (0.99 to 1.59) | .056 | 1.14 (0.90 to 1.44) | .285 |
| 2, N = 1,985 | 2.13 (1.69 to 2.69) | < .001 | 1.86 (1.47 to 2.36) | < .001 |
| 3, N = 994 | 3.39 (2.64 to 4.36) | < .001 | 2.76 (2.13 to 3.57) | < .001 |
| ≥4, N = 567 | 4.51 (3.38 to 6.02) | < .001 | 3.53 (2.62 to 4.76) | < .001 |
| Linear effect of load | 1.46 (1.38 to 1.55) | < .001 | 1.39 (1.31 to 1.47) | < .001 |

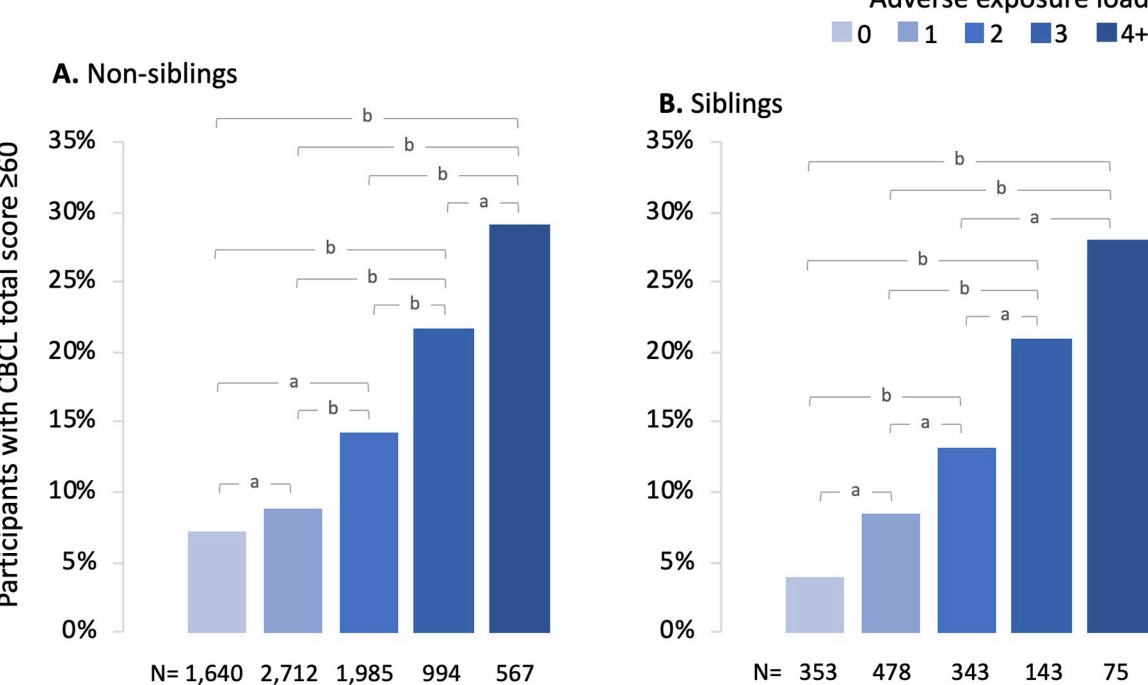

**Fig 1. Frequency of CBCL total score above the normal range (≥60) as related to adverse prenatal exposure load.** a, p < .05; b, p < .001 for pairwise differences in odds of CBCL total ≥60 in fully adjusted models.

Siblings within families differed in exposure load by 0 to 4 exposures. Greater discordance between sibling pairs associated with a greater difference in CBCL total score, after controlling for other variables that could differ between siblings (age, sex, maternal age at birth) and mean family exposure load, with a difference of 0.67 CBCL points for each level of discordance (95% CI 0.12 to 1.22, p = .017, S2 Fig in S1 File). After including these within-family effects in the model, the between-family effects became non-significant (p = .077).

## Discussion

In this study of 9,290 children in the ABCD cohort, loading of adverse prenatal exposures associated with linear increases in psychopathology at age 9–10, in both dimensional and clinically thresholded models. Nearly identical effects of exposure load were observed in two groups of children studied in parallel–those with versus without a sibling enrolled. Across five exposures–unplanned pregnancy, maternal use of alcohol or tobacco early in pregnancy, and obstetric complications during pregnancy or at birth–that each independently associated with increased CBCL total scores, a combination of two or more such exposures associated with significantly increased odds of clinically meaningful psychopathology (CBCL total score ≥60). Such loading was a common occurrence in the study population, as 44% of children had at least two exposures. A combination of four our more exposure increased odds of clinically meaningful psychopathology by approximately 4-fold. Post hoc analyses of individual syndrome and broad spectrum CBCL scales yielded analogous effects of exposure load across all domains of psychopathology. These differences survived control for numerous postnatal factors that impact psychopathology risk, including those at the level of families, neighborhoods, and study sites. In particular, among siblings discordant for exposure load, higher CBCL scores were observed among siblings with higher exposure load. Despite the smaller sample size for

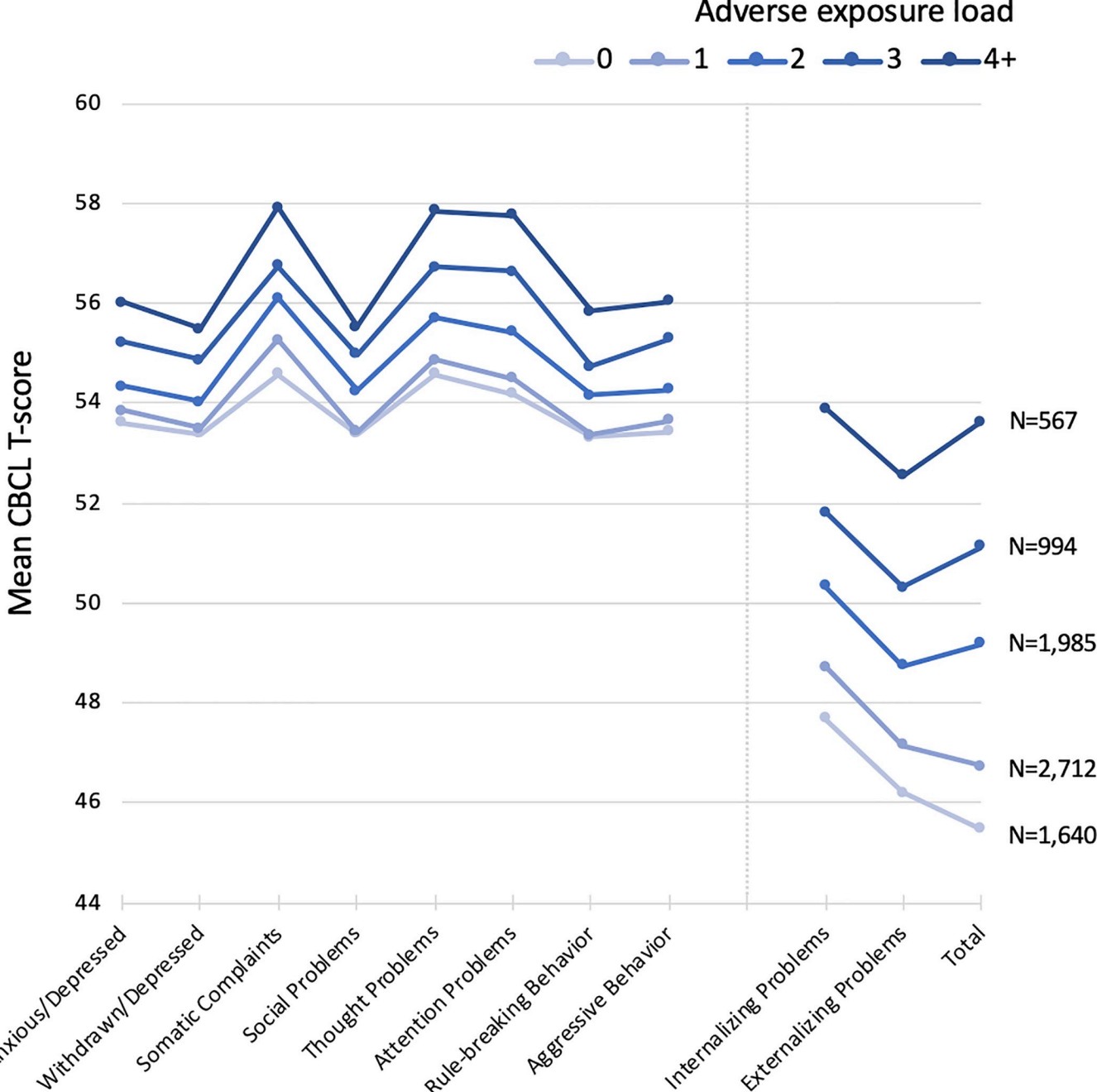

**Fig 2. Increased adverse exposure load associated with higher individual syndrome and broad-band CBCL T-scores in the non-sibling group.** The association was significant for each T-score scale in fully adjusted models (p < .05, FDR corrected). T-scores ≥65 for individual syndrome scores, and ≥60 for broad-band scores, are considered clinically significant.

that analysis, this finding suggests that the relationship between prenatal exposure load and CBCL is unlikely to be confounded by unmeasured family-level differences, a possibility suggested by previous work (e.g., [19, 51]).

We included in the full model a range of covariates, including those that could counfound relationships between prenatal exposures and CBCL scores (e.g., socioeconomic factors). In sensitivity analyses, we also included exposures that could not directly predispose to events

**Table 3. Effect of adverse prenatal exposure load on odds of CBCL total score ≥60 in the validation (sibling) sample.**

| EXPOSURE LOAD, N | Odds of CBCL total ≥60 (minimally adjusted) | | Odds of CBCL total ≥60 (fully adjusted) | |
|---|---|---|---|---|
| | Odds ratio (95% CI) | p | Odds ratio (95% CI) | p |
| 0, N = 353 | Reference | – | Reference | – |
| 1, N = 478 | 3.17 (0.74 to 13.51) | .118 | 1.92 (0.93 to 3.96) | .077 |
| 2, N = 343 | 3.82 (0.85 to 17.06) | .079 | 2.97 (1.41 to 6.25) | .004 |
| 3, N = 143 | 5.76 (1.06 to 31.48) | .043 | 4.96 (2.07 to 11.87) | < .001 |
| ≥4, N = 75 | 10.10 (1.24 to 82.27) | .031 | 6.88 (2.26 to 20.94) | .001 |
| Linear effect of load | 1.64 (1.44 to 2.35) | .007 | 1.61 (1.30 to 2.01) | < .001 |

during pregnancy but are known to associate with CBCL (e.g., early childhood trauma); of note, these variables could also potentially be affected by CBCL scores (e.g., screen time, family conflict). This approach comes at the potential cost of attenuating causal effects of the adverse prenatal exposures, if, for example, some of the included covariates mediate their relationship to child psychopathology. However, the effects of exposure burden on CBCL scores were comparable regardless of the number of included covariates, as seen by comparing the minimally and fully adjusted models to those in sensitivity analyses. This pattern suggests that the relationship between exposure load and child psychopathology is robust, regardless of the complex effects that the covariates likely confer.

A growing literature has related adverse prenatal exposures to risk for subsequent psychopathology. While each of the exposures implicated herein have associated with psychopathology risk in previous studies, almost all have considered them in isolation, and many have related individual exposures to specific diagnoses or syndromes. For example, numerous studies have associated prenatal alcohol [13, 14, 17] or tobacco [15, 16] exposure with risk for conduct disorder or other externalizing symptoms. Studies of obstetric complications are weighted toward severe mental illnesses such as schizophrenia and bipolar disorder [9], although some have considered their relationship to psychopathology more broadly [32, 34]. There have been fewer studies of unplanned or unwanted pregnancies and offspring mental health, although several have found associations with externalizing psychopathology [52] and psychosis [53, 54]. There also remain questions about whether the timing of exposure (i.e., early versus late in pregnancy) matters in regard to risk for psychopathology [55, 56].

With its large and systematically characterized cohort, the ABCD Study provides the opportunity to disentangle relationships between a range of exposures and different clinical outcomes. The CBCL is a versatile instrument as it assesses numerous categories of psychopathology that can be assessed either dimensionally or categorically. Here, we first used dimensional assessment to detect subtle relationships between exposures and clinical syndromes across a large population, most of whom do not meet the threshold of illness. Once these prenatal factors were identified, we used threshold-based models to quantify risk for clinically relevant psychopathology. In so doing we found that the relationship between prenatal exposure load and psychopathology risk was less complex than might have been imagined in light of previous, more narrowly focused studies: more exposures, whether occurring early or late in pregnancy, associate with greater risk for a broad spectrum of psychopathology.

This pattern is intriguing because it suggests the possibility of common biological pathways through which prenatal insults may influence risk for a variety of psychiatric symptoms. For example, recent work has implicated the placenta in mediating the relationship between obstetric complications and offspring risk for serious mental illness [57, 58]. While the present findings do not offer any specific mechanistic insights, additional translational work in this area might identify targets for early interventions that mitigate risk across a range of

psychiatric syndromes. In the near term, though, quantitative assessments of cumulative prenatal risk such as the one described here may identify children who could benefit from promising early psychosocial interventions, such as enriched learning environments, that may temper risk conferred by adverse prenatal exposures [59–61].

In regard to the present analysis, a principal limitation of the ABCD Study design is its reliance on retrospective self-report of prenatal exposures. While reported exposure rates here were similar to those obtained in cross-sectional national studies of rates complicated pregnancy (40% here vs. 47% [62]), unplanned pregnancy (40% vs. 33% [63]), birth complications (24% vs. 20% [64]), and early tobacco use (14% vs. 15% [65]), the present design may have introduced recall error and bias that a fully prospective study with objective reporting from medical records could minimize [66]. While some factors (e.g., obstetric complications) have been shown to be robust to long-term recall in other studies [62, 64, 67, 68], other factors vary in their reliability for complex reasons that could over- or underestimate the results. For example, substance use during pregnancy is commonly underreported, an effect that would tend to diminish the strength of the present findings; one recent study found that 61% of women endorsed alcohol consumption between conception and recognition of pregnancy [69], more than twice the rate reported here. However, it is notable several prospective studies of substance exposure during pregnancy report smaller or null associations with childhood psychopathology [19, 70, 71]. Unplanned pregnancy has been shown to be both over- and underestimated on the order of 20% [72, 73]. Accordingly, the measurement error related to these factors in the present study is likely significant, but not necessarily biased in one direction, and in some cases may pull toward the null hypothesis. The large and diverse sample afforded by the ABCD study appears to have been sufficient for signal detection, even after controlling for numerous potential confounders.

Of note, study participants were born before the current opioid epidemic, and opiate exposure during pregnancy occurred with insufficient frequency to study. To be consistent with prior studies and facilitate a comparison of results, substance use was treated as a binary variable, as were obstetric complications, although their effects on child psychopathology may be dose-dependent [74]. Other prenatal factors that are known to influence psychopathology risk, including maternal infection [11, 12], nutrition [75, 76], stressful events [52] and mental health [77–79], were not measured in the ABCD study; as such the adversity loading model presented here, while accounting for substantially more variance in risk than individual factors, is certainly incomplete. These findings may provide proof-of-concept for fully prospective longitudinal studies that thoroughly canvass the prenatal environment. Finally, the included exposures likely have heritable and non-heritable components; exposures that are more heavily influenced by heritable factors are potentially more exposed to family-level confounding. However, the relatively small effect of unmeasured family-level confounders within the sibling participants might lessen this concern.

These limitations should be weighed against the many strengths of the ABCD Study: a cohort that is large, racially and socioeconomically diverse, relatively homogenous with regard to age, and uniformly characterized; inclusion of a range of potential postnatal exposures that are known to influence psychopathology; and the opportunity to control for other unmeasured, family-level confounders through replication in a sibling cohort. Sibling designs such as these control for systematic maternal rating bias, although they do not fully adjust for child-parent effects. The sample is geographically diverse and reflects the socioeconomic, racial, and ethnic diversity of the U.S., an advantage compared to other fully prospective but single-city cohorts (ALSPAC, Generation R); notably, though, as the ABCD study recruited from academic centers, urban/suburban populations are better represented, and results may not generalize as well to rural communities. Perhaps most importantly, the ABCD Study provides an

opportunity to follow these participants through the high-risk adolescent years, enabling future study of the stability of these patterns, their relationship to emergent psychopathology and brain development, and their interaction with adverse life events that are captured prospectively.

In summary, leveraging data from the ABCD Study, we report 5 prenatal exposures that independently and additively associate with dimensional measures of psychopathology at age 9–10. The cumulative impact of these factors is clinically relevant and reproducible, and appears robust to potential postnatal and familial confounders, although bias cannot be ruled out in this retrospective design. These findings support the call for fully prospective, broadly representative studies that begin during pregnancy, such as the proposed NIH HEALthy Brain and Child Development Study. They also underscore the importance of policies and interventions to promote healthy pregnancies as a means of protecting the offspring's brain health in childhood.

## Supporting information

**S1 File.**
(DOCX)

## Author Contributions

**Conceptualization:** Joshua L. Roffman, Alysa E. Doyle, Erin C. Dunn.

**Data curation:** Joshua L. Roffman, Eren D. Sipahi, Kevin F. Dowling, Dylan E. Hughes, Casey E. Hopkinson.

**Formal analysis:** Joshua L. Roffman, Eren D. Sipahi, Kevin F. Dowling, Dylan E. Hughes, Casey E. Hopkinson, Hang Lee.

**Funding acquisition:** Joshua L. Roffman, Erin C. Dunn.

**Investigation:** Joshua L. Roffman, Hamdi Eryilmaz, Lee S. Cohen, Jodi Gilman, Alysa E. Doyle, Erin C. Dunn.

**Methodology:** Joshua L. Roffman, Erin C. Dunn.

**Supervision:** Joshua L. Roffman, Alysa E. Doyle, Erin C. Dunn.

**Writing – original draft:** Joshua L. Roffman, Eren D. Sipahi, Kevin F. Dowling, Dylan E. Hughes, Casey E. Hopkinson, Hang Lee, Hamdi Eryilmaz, Lee S. Cohen, Jodi Gilman, Alysa E. Doyle, Erin C. Dunn.

**Writing – review & editing:** Joshua L. Roffman, Eren D. Sipahi, Kevin F. Dowling, Dylan E. Hughes, Casey E. Hopkinson, Hang Lee, Hamdi Eryilmaz, Lee S. Cohen, Jodi Gilman, Alysa E. Doyle, Erin C. Dunn.

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
