## [Decision Letter · Decision Letter 0]

14 Aug 2020

PONE-D-20-15151

Association of adverse prenatal exposure burden with dimensional child psychopathology in the Adolescent Brain Cognitive Development (ABCD) Study

PLOS ONE

Dear Dr. Roffman,

Thank you for submitting your manuscript to PLOS ONE. After careful consideration, we feel that it has merit but does not fully meet PLOS ONE’s publication criteria as it currently stands. Therefore, we invite you to submit a revised version of the manuscript that addresses the points raised during the review process.

We look forward to receiving your revised manuscript.

Kind regards,

Maria Christine Magnus, MPH

Academic Editor

PLOS ONE

Journal Requirements:

"Institutional review board approval was obtained at each site. All parents provided written informed consent and all children provided assent."

b) Once you have amended this statement in the Methods section of the manuscript, please add the same text to the “Ethics Statement” field of the submission form (via “Edit Submission”).

Reviewers' comments:

Reviewer's Responses to Questions

**Comments to the Author**

1. Is the manuscript technically sound, and do the data support the conclusions?

Reviewer #1: Yes

Reviewer #2: Partly

2. Has the statistical analysis been performed appropriately and rigorously? 

Reviewer #1: I Don't Know

Reviewer #2: No

3. Have the authors made all data underlying the findings in their manuscript fully available?

Reviewer #1: Yes

Reviewer #2: Yes

4. Is the manuscript presented in an intelligible fashion and written in standard English?

Reviewer #1: Yes

Reviewer #2: Yes

5. Review Comments to the Author

Reviewer #1: In this study of 9,290 children in the ABCD cohort, cumulative loading of adverse prenatal exposures associated with increased odds of psychopathology symptoms that fall within the clinically significant range, even after sibling control. Exploring the cumulative effect of prenatal exposures are highly interesting. However, there are some weaknesses:

1) The reliance on retrospective self-report of prenatal exposures is a major limitation in this study. This is of course nothing the authors can do anything to change, and the limitation is underlined in the discussion.

2) The use of a sibling control design is a major strength of this study. However, I do not think the design and the results of it receive enough space in the text. In the current version, it is barely mentioned in the abstract. In the introduction I would recommend to include a short introduction to sibling designs, and a description of why this design represent a strength compared to other association studies. Previous sibling studies should also be referred to. In the result section, I would prefer to first see results without sibling control (only accounting for independence in the data) – these should be compared to the results from the discovery sample. And next, see how much of the associations that were explained by the unmeasured family confounding. The power of a sibling design increase when using a dimensional outcome. I would report effects of cumulative exposure on both dimensional and binary outcomes after sibling control.

3) Aims: I would suggest the first aim to be to investigate the cumulative effect of adverse prenatal exposures on the dimensional outcome. And next, the cumulative effect on the binary outcome. Both of the above should be run in the sibling sample, adjusting for unmeasured confounding.

4) Selection of exposures. Why did you exclude exposures present in less than 5% pof the sample? They could play a role in the dose-response framework. On the other hand, some of the exposures are binary variables calculated based on several variables (e.g. pregnancy and birth complications). I guess some of the children will have experienced several complications. Why were not all the single exposures counted to make up the cumulative variable? In other cases, variables were removed instead of being combined with others (e.g. marihuana use could be combined with smoking?). I would prefer a consistent pattern of selection.

Another point – the selected exposures represent both factors that are related to maternal heritable traits of mental health and personality (e.g. unplanned pregnancy, alcohol use in pregnancy) and more random factors (e.g. birth and pregnancy complications). I would assume that unmeasured familial confounding to explain more of associations with the first group of exposures..

5) Selection of covariates – It seems that some of the covariates selected are related to the present situation -at the time the child is 9-10 years (for example the presence or absence of a partner, screentime, family conflict, etc.). Even though these phenotypes are associated with the exposures, they could not be causing them (since they appear after the exposure in time). Rather, this adjustment could lead to over-adjustment. This is also the case when adjusting for alcohol use and smoking after pregnancy. It is likely that people smoking and drinking during pregnancy would continue to do this after pregnancy as well. By adjusting for these variables, a real effect between smoking/drinking during pregnancy would be removed. It would be interesting to see results without these adjustments as well. It would be much better to have a measure of maternal history of mental health before pregnancy, but I guess that is not available.

Minor comments:

Title/Abstract:

• I would remove the word dimensional from the title (and from the objective in the abstract (line 52). As long as you state in the methods section that you use a dimensional measure of psychopathology. In addition, you also run analyses with a cut-off.

• Report betas/ORs with CIs instead of p-values. I would also recommend focusing on the results of the main aim – the cumulative effect of the exposures on dimensional outcomes (in discovery and sibling sample), and also the cumulative effect of the exposures on the binary outcome. (not on the independent effects of the various exposures).

Introduction:

• Line 101 - The analogy of polygenic risk scores should be rewritten if included. Right now it is not clear to me what the message is. I guess the message is that it seems that bot environmental and genetic influences are made up of the additive influence of very small influences dispersed across the genome/phenome.

• Possible mechanisms should be described in the introduction. Why would we expect a dose-response relationship?

• Line 178 -Prenatal exposures – Under this heading 5 late exposures are also described. These could be removed from this description.

• Why did you include only 5 imputation sets? I thought a higher number was more often used.

Reviewer #2: This is a study of potential interest, but as of now still contains major and moderate theoretical and methodological weaknesses. The major uncircumventable weakness is of course the retrospective report on the pregnancy. The introduction and discussion shold make clear which prior studies have prospective and retrospective reporting.

In the introduction the authors allude to common causal pathways, but for some reason study independent contributions of risk factors. This does not make sense. The authors should start out with an analysis leading to aggregation of risk factors and consider using techniques such as lasso regression and elastic net to do this. It is also paramount that the authors include a DAG of the selected covariates in the model to be clear on the hypothesized links between them internally and to the outcome. Some of the covariates (potential consequences of child psychopathology) can potentially introduce collider stratification bias. Other covariates are potential mediators; these would attenuate the estimate of the causal effect of the prenatal risk factors (like adjusting smoking for nicotine dependence when studying lung cancer).

Introduction

• Speculative to relate these subjects to Barker, Hunger Winter and Chinese famine. The mechanisms are very much unknown and not evaluated in this study. The potential mechanisms could be indefinite, but these are mostly highlighted because they are currently famous, not because they are the most likely model for these data.

• Same goes for prematurity, maternal infections.

• Cite multivariate twin, family, and genomic studies on substance abuse and psychopathology.

• The literature seems somewhat selected to support the current study, but associations are confounded with contrafactuals. Please separate the prior findings into association studies and the much smaller number of contrafactual studies (e.g. sibling designs, children of twins, IVF, and mendelian randomization). This would also bring to light the novelty of the current sibling study.

• The term pluripotent is ambiguous in this context. The authors should state more cleary what their hypotheses are and if they are also referring to non-additive/interaction effects of exposures.

• Line 138: The term increased dimensional psychopathology is uncelar.

• The authors state that “multiple individual adverse prenatal exposures would associate independently with increased dimensional psychopathology”. They should include in the manuscript a DAG of all these hypothesized associations. Hypotheses about causal links between the chosen covariates are of great importance when interpreting the independent contributions.

Methods

• The term “discovery sample” does not fit well here. The sibling and non-sibling samples represent to some extent different parts of the population. Total effects should therefore also be estimated in the sibling sample.

• Measures of screen time, family conflict, and some significant traumas could be consequences of the outcome (e.g. by evocative or active gene-environment correlations) leading to a potential collider stratification bias. All covariates that are potential consequences of child psychopathology should be omitted from the analyses.

• Variables that are potential mediators should be only included in additional sensitivity analyses. (akin to adjusting the smoking – lung cancer association for nicotine dependence; leading to null effects).

• The authors should present more detailed information on the extent of missing data. Regardless, five imputations is most likely far from enough. Run with as many imputations as your machinery can perform (e.g. 50, 100, or 200 imputions).

• The imputation analysis must be informed on the multilevel structure of the data. This is a major concern. The cites can be imputed separately (some applications have a “cluster” feature for this), and siblings can be imputed in a wide format and the flipped back to a long format.

o Audigier, V., White, I. R., Jolani, S., Debray, T. P., Quartagno, M., Carpenter, J., ... & Resche-Rigon, M. (2018). Multiple imputation for multilevel data with continuous and binary variables. Statistical Science, 33(2), 160-183.

o Grund, S., Lüdtke, O., & Robitzsch, A. (2018). Multiple imputation of missing data for multilevel models: Simulations and recommendations. Organizational Research Methods, 21(1), 111-149.

o https://bookdown.org/mwheymans/bookmi/multiple-imputation-models-for-multilevel-data.html

• The authors should describe the type of multiple imputation used (e.g. multivariate normal).

• Are there sampling weights available for the ABCD study? If so, the authors should use them.

• There is a large number of covariates used. The authors could consider using lasso or elastic net regression to simplify the model (e.g. available in STATA 16). Such approaches inform on the extent of out of sample replicability (such as the sibling sample).

• It is not necessary to state the variance component type when having only a random intercept (cite). 21 cites is not many. The authors could also consider accounting for cite effects using fixed effects (i.e. adjusting for them by dummies). This does not presume a normally distributed latent “cite factor”. This opens the door to do all kinds of analyses where multi level models are not easily available (e.g. elastic net).

• The calculation of the “integer sum of a child’s prenatal exposures” was guided by the independent contributions. This seems absurdly misguided. In the introduction the authors describe typical common pathway mechanisms for exposures. This is akin to predicting depression using the NEO-PI-R neuroticism items, but only basing the depression risk score on the items that independently predicting depression. The point here is to aggregate risk factors, not disaggregate. One blunt option is to do bivariate analyses and then calculate a risk score based on these associations. I would strongly recommend that the authors apply a sound variable selection procedure such as lasso or elastic net regression to create the risk score.

• The authors use the term “clinically significant psychopathology” for >65 on CBCL syndrome scales and >60 for CBCL “broad band scales” (standard term?). According to the CBCL manual these are the threshold for “borderline clinical range”. Please revise the text accordingly. The “clinical range” is according to the manual set at a higher level.

• Please also include linear models with the CBCL continuous scores for comparison. This would be especially helpful in the sibling analyses, since dichotomization introduces measuring error; introducing random differences between siblings close to the threshold.

• If all siblings were assessed at the same site, it is not necessary to introduce site as a random effect. They are part of the between sibling effects. You could rather just use a sandwich estimator to correct the standard errors for cite dependencies.

• Please cluster-mean center the exposure variables and present both effects of cluster mean and the centered individual variables (keep them both in the analysis). In this way covariates at both level 1 and 2 are interpretable. The reader can then assess the familial (between) risk independently from the within family risk. (i.e. what is the risk of descending from a high-risk family).

Discussion

• The authors should in depth discuss to what extent their results are a result of retrospective reporting. What is more discuss why sibling studies using prospective designs to a lesser or no extent find an association between substance use and distress during pregnancy and child psychopathology (e.g (Eilertsen et al., 2017; Hannigan et al., 2018; Lund et al., 2019), but do find this for postnatal stress (e.g. (Gjerde et al., 2019; Lund et al., 2020). As it is now it appears that the authors have cherry-picked literature supporting their findings, when in reality the small literature using contrafactual studies goes in different directions.

• Although the sibling design serves as a natural experiment for exposures during pregnancy (mothers share 100% of their own genome across pregnancies), the design does not fully adjust for child-parent effects (full siblings only share 50% of their genome). This should be made clear to the reader.

• The authors should also bring to light the strengths of the sibling design in that it cancels systematic maternal rating bias.

Eilertsen, E. M., Gjerde, L. C., Reichborn-Kjennerud, T., Orstavik, R. E., Knudsen, G. P., Stoltenberg, C., . . . Ystrom, E. (2017). Maternal alcohol use during pregnancy and offspring attention-deficit hyperactivity disorder (ADHD): a prospective sibling control study. International Journal of Epidemiology, 46(5), 1633-1640. doi:10.1093/ije/dyx067

Gjerde, L. C., Eilertsen, E. M., Hannigan, L. J., Eley, T., Roysamb, E., Reichborn-Kjennerud, T., . . . Ystrom, E. (2019). Associations between maternal depressive symptoms and risk for offspring early-life psychopathology: the role of genetic and non-genetic mechanisms. Psychological Medicine, 1-9. doi:10.1017/S0033291719003301

Hannigan, L. J., Eilertsen, E. M., Gjerde, L. C., Reichborn-Kjennerud, T., Eley, T. C., Rijsdijk, F. V., . . . McAdams, T. A. (2018). Maternal prenatal depressive symptoms and risk for early-life psychopathology in offspring: genetic analyses in the Norwegian Mother and Child Birth Cohort Study. Lancet Psychiatry, 5(10), 808-815. doi:10.1016/S2215-0366(18)30225-6

Lund, I. O., Moen Eilertsen, E., Gjerde, L. C., Ask Torvik, F., Roysamb, E., Reichborn-Kjennerud, T., & Ystrom, E. (2020). Maternal Drinking and Child Emotional and Behavior Problems. Pediatrics, 145(3). doi:10.1542/peds.2019-2007

Lund, I. O., Moen Eilertsen, E., Gjerde, L. C., Roysamb, E., Wood, M., Reichborn-Kjennerud, T., & Ystrom, E. (2019). Is the association between maternal alcohol consumption in pregnancy and pre-school child behavioural and emotional problems causal? Multiple approaches for controlling unmeasured confounding. Addiction, 114(6), 1004-1014. doi:10.1111/add.14573

6. PLOS authors have the option to publish the peer review history of their article (what does this mean?). If published, this will include your full peer review and any attached files.

Reviewer #1: No

Reviewer #2: **Yes: **Eivind Ystrom

---

## [Author Response · Author response to Decision Letter 0]

1 Feb 2021

NB: PLEASE REFER TO ATTACHED RESPONSE TO REVIEW DOCUMENT FOR FULLY FORMATTED VERSION, INCLUDING LEGIBLE TABLES

We are grateful for the comments of the editor and reviewer, to which we respond below point-by-point.

Journal Requirements

Done.

2) Thank you for including your ethics statement:

"Institutional review board approval was obtained at each site. All parents provided written informed consent and all children provided assent."

We have amended the ethics statement as follows (p.9):

“IRB approval for the ABCD study is described in Auchter et al. (46). Most ABCD research sites cede approval to a central Institutional Review Board (cIRB) at the University of California, San Diego, with the remainder obtaining local IRB approval. All parents provided written informed consent and all children provided assent.”

b) Once you have amended this statement in the Methods section of the manuscript, please add the same text to the “Ethics Statement” field of the submission form (via “Edit Submission”).

We have uploaded the amended ethics statement as above.

Reviewer #1

1) The reliance on retrospective self-report of prenatal exposures is a major limitation in this study. This is of course nothing the authors can do anything to change, and the limitation is underlined in the discussion.

We agree with the reviewer that this is an important but inevitable limitation of this data set, as recall error and bias are possible. That rates of obstetric exposures are similar to those reported in other large, prospective studies provides some reassurance (p.22). We hope that the results of this study provide impetus for additional, fully prospective investigations of cumulative effects of adverse prenatal exposures on risk for psychopathology in children. 

2) The use of a sibling control design is a major strength of this study. However, I do not think the design and the results of it receive enough space in the text. In the current version, it is barely mentioned in the abstract. In the introduction I would recommend to include a short introduction to sibling designs, and a description of why this design represent a strength compared to other association studies. Previous sibling studies should also be referred to. In the result section, I would prefer to first see results without sibling control (only accounting for independence in the data) – these should be compared to the results from the discovery sample. And next, see how much of the associations that were explained by the unmeasured family confounding. The power of a sibling design increase when using a dimensional outcome. I would report effects of cumulative exposure on both dimensional and binary outcomes after sibling control.

Thank you for this suggestion, which provides the opportunity to characterize the potential impact of unmeasured family-level confounders in greater detail. We have made several changes to the manuscript in accordance with the reviewer’s recommendations:

- Abstract/Results: we added “Within sibling pairs, greater discordance for exposure load associated with greater CBCL total differences, suggesting that results in the non-sibling study were not confounded by unmeasured family-level effects.” (p.3)

- Introduction: we added “The ABCD Study provides extensive phenotyping of both prenatal and postnatal exposures that have associated with psychopathology risk (e.g., trauma, family conflict) in prior studies (36-39), as well as of dozens of demographic and environmental features that potentially confound these relationships. The ABCD Study also includes siblings, and analysis of sibling pairs enables additional control over potential confounding effects of unmeasured individual- and family-level variables. Specifically, siblings who are discordant for exposures should show greater differences in psychopathology, a pattern that has supported causal relationships in studies of prenatal exposure to tobacco (40), alcohol (41), and obstetrical complications (42).” (p.6)

- Results: we added additional tables from the Sibling data that include (a) effects of exposure load on CBCL total as a continuous measure, without controlling for family-level effects (Table S11):

EXPOSURE LOAD, N Effect on CBCL Total

(minimally adjusted) Effect on CBCL Total

(fully adjusted)

 Estimate (95% CI) p Estimate (95% CI) p

0, N=357 Reference -- Reference --

1, N=479 2.12 (0.60 to 3.65) .006 1.49 (0.08 to 2.90) .038

2, N=356 4.85 (3.22 to 6.50) <.001 3.36 (1.82 to 4.90) <.001

3, N=146 8.50 (6.26 to 10.75) <.001 6.21 (4.10 to 8.32) <.001

≥4, N=54 9.26 (5.82 to 12.69) <.001 6.39 (3.20 to 9.57) <.001

(b) effects of exposure load on continuous CBCL total scores with controlling for family-level effects (Table S10):

EXPOSURE LOAD, N Effect on CBCL Total

(minimally adjusted) Effect on CBCL Total 

(fully adjusted)

 Estimate (95% CI) p Estimate (95% CI) p

0, N=357 Reference -- Reference --

1, N=479 1.71 (0.30 to 3.12) .018 1.29 (-0.06 to 2.65) .062

2, N=356 3.67 (2.09 to 5.25) <.001 2.70 (1.18 to 4.22) <.001

3, N=146 6.68 (4.55 to 8.81) <.001 5.11 (3.05 to 7.18) <.001

≥4, N=54 8.32 (5.06 to 11.59) <.001 6.09 (2.96 to 9.22) <.001

(c) effects of exposure load on binary CBCL total scores (<60 vs ≥60) without controlling for family-level effects, Table S12):

EXPOSURE LOAD, N Odds of CBCL total ≥60 

(minimally adjusted) Odds of CBCL total ≥60 

(fully adjusted)

 Odds ratio (95% CI) p Odds ratio (95% CI) p

0, N=357 Reference -- Reference --

1, N=479 2.15 (1.13 to 4.08) 0.019 1.94 (0.99 to 3.81) .054

2, N=356 3.69 (1.96 to 6.93) <.001 2.76 (1.41 to 5.41) .003

3, N=146 6.68 (3.30 to 13.51) <.001 4.27 (1.98 to 9.21) <.001

≥4, N=54 6.85 (2.76 to 16.98) <.001 4.00 (1.50 to 10.66) .006

For comparison, here are effects of exposure load on binary CBCL total scores (<60 vs ≥60) with controlling for family-level effects (Table 3, as in the original version, but now updated to reflect 200 imputation sets instead of 5):

EXPOSURE LOAD, N Odds of CBCL total ≥60 

(minimally adjusted) Odds of CBCL total ≥60 

(fully adjusted)

 Odds ratio (95% CI) p Odds ratio (95% CI) p

0, N=357 Reference -- Reference --

1, N=479 3.34 (0.75 to 14.80) .112 1.78 (0.86 to 3.65) .118

2, N=356 4.10 (0.80 to 20.99) .090 2.48 (1.19 to 5.16) .015

3, N=146 8.53 (1.37 to 53.31) .022 3.92 (1.68 to 9.14) .002

≥4, N=54 20.71 (2.15 to 199.25) .009 3.90 (1.26 to 12.08) .018

Note that for both continuous and dichotomous CBCL outcome measures, within the fully adjusted models, including family-level effects had a fairly minimal impact and does not change the interpretation of results. 

Relatedly, we added the following text to the Results section: “Further, we repeated the analysis without including family ID as a random effect to determine the extent that unmeasured family confounding influenced the results. Resultant odds ratios and confidence intervals were similar (Tables S11, S12) suggesting that effects of family-level confounders were minimal.” (p.18) 

Finally, we appreciate the reviewer’s suggestion to compare dichotomous CBCL outcomes (<60 vs ≥60) between discordant sibs. However, the number of sib pairs with sufficient discordance for prenatal exposure load (i.e., discordance level differing by at least 2 exposures, per Tables 2 and 3) as well as discordance for CBCL category (i.e., one sib <60 and the other sib ≥60) was quite small, comprising only 21 pairs. Within these pairs, the sib with the greater exposure load was more likely to be the one with the CBCL score ≥60, with an odds ratio similar to that seen in the non-sibling group (1.78); but in a sample this small, such an effect size does not approach statistical significance (95% CI 0.52 to 6.04). As such we are likely underpowered for this analysis, and have elected not to include it in the manuscript; indeed, as the reviewer emphasized, sibling design studies are better powered for continuous outcomes, as was the case here.

3) Aims: I would suggest the first aim to be to investigate the cumulative effect of adverse prenatal exposures on the dimensional outcome. And next, the cumulative effect on the binary outcome. Both of the above should be run in the sibling sample, adjusting for unmeasured confounding.

Thank you for the suggestion. We made the following changes along these lines:

- Abstract/Methods: we added “We also assessed cumulative effects of these factors CBCL total as a continuous measure, as well as on odds of clinically significant psychopathology (CBCL total ≥60).” (p.2)

- Introduction: we added “We hypothesized that [1] multiple individual adverse prenatal exposures would associate independently with increased dimensional psychopathology; and [2] increased loading for such individual adverse prenatal exposures would associate with greater odds of psychopathology, as measured both dimensionally and via thresholded indices of clinically-relevant psychopathology.” (p.7)

- We now report cumulative effects of adverse prenatal exposures on continuous CBCL outcomes, to complement the dichotomous outcome analyses. For the non-sibling group, this is included in Table S8:

EXPOSURE LOAD, N Effect on CBCL Total

(minimally adjusted) Effect on CBCL Total

(fully adjusted)

 Estimate (95% CI) p Estimate (95% CI) p

0, N=1,650 Reference -- Reference --

1, N=2,760 1.54 (0.86 to 2.21) <.001 0.95 (0.32 to 1.59) <.001

2, N=2,079 4.25 (3.53 to 4.98) <.001 2.88 (2.18 to 3.57) <.001

3, N=986 6.44 (5.53 to 7.35) <.001 4.50 (3.63 to 5.38) <.001

≥4, N=423 8.83 (7.55 to 10.11) <.001 6.26 (5.03 to 7.49) <.001

Linear effect of load 2.20 (1.97 to 2.44) <.001 1.55 (1.32 to 1.78) <.001

For the sibling group, it is included in Table S10 (see response to Reviewer 1, Point 2 above).

4) Selection of exposures. Why did you exclude exposures present in less than 5% of the sample? They could play a role in the dose-response framework. On the other hand, some of the exposures are binary variables calculated based on several variables (e.g. pregnancy and birth complications). I guess some of the children will have experienced several complications. Why were not all the single exposures counted to make up the cumulative variable? In other cases, variables were removed instead of being combined with others (e.g. marihuana use could be combined with smoking?). I would prefer a consistent pattern of selection.

We appreciate the opportunity to clarify our approach and have revised the manuscript accordingly.

- Exposures were grouped based on the approach of previous studies of prenatal exposures and psychopathology risk. We sought to adhere as closely as possible to these precedents, so that the current results (and the contrast between effects of individual vs. cumulative exposures) were interpretable within this established context. For example, we grouped all pregnancy complications – regardless of the number of complications -- together under one term, as did Laurens et al., BMC Psychiatry, 2015. In contrast, we did not condense marijuana and tobacco smoking into one variable because previous studies (e.g., Paul et al., JAMA Psychiatry, 2020 and Talati et al., Psychiatry Res, 2017, respectively) had linked each one (separately) to altered neurodevelopment or psychopathology risk. Of note, exposure by category, as opposed to number of events within each category, is an approach that is frequently used in the adverse childhood experiences (ACE) literature (e.g., Lansing et al., J Anxiety Dis 2017; Merrick MT et al., Child Abuse Negl 2017).

- We only included prenatal exposures in the cumulative analysis if (1) they were present in at least 5% of the non-sibling sample, and (2) within that sample they were independently associated with risk of increased CBCL scores. Exposures present in less than 5% (i.e., <395 participants) were not included as exposures of interest to avoid potential overfitting. As a “rule of thumb” at least 10-15 participants are required for each covariate entered into the model (e.g., Harrell FE, Regression modeling strategies, Springer-Verlag, 2001; Schmidt FL, Educ Psychol Meas, 1971). Below the 5% threshold, with 30 total covariates included in the model, the subjects per variable are approximately 13 or fewer. Accordingly, when examining cumulative exposure effects, we only included those individual exposures that we could confidently relate (independently) to the psychopathology outcome. 

- Accordingly, we added the following text to Methods/Prenatal Exposures: 

“The following 15 exposures were extracted for each individual and coded as present or absent, consistent with dichotomous analyses used in previous studies (e.g., (9, 18, 25, 26)). (p.9) 

“The 5% threshold was chosen a priori to minimize the risk of overfitting in smaller groups, given the total number of covariates.” (p.10)

- We also added the following text to the Discussion:

“To be consistent with prior studies and facilitate a comparison of results, substance use was treated as a binary variable, as were obstetric complications, although in the effects on child psychopathology may be dose-dependent (74).” (p.23)

Another point – the selected exposures represent both factors that are related to maternal heritable traits of mental health and personality (e.g. unplanned pregnancy, alcohol use in pregnancy) and more random factors (e.g. birth and pregnancy complications). I would assume that unmeasured familial confounding to explain more of associations with the first group of exposures.

Thanks for this thoughtful point. Parsing heritable versus non-heritable aspects of prenatal risk factors is challenging, and early studies have yielded some surprising results (e.g., obstetric complications not only are not random, but share overlapping genetic risk with schizophrenia risk; see Ursini et al., Nat Med 2018). Given that familial confounding did not substantially influence effects of these risk factors on CBCL outcomes, we are less concerned effects of heritable vs. non-heritable factors on exposure risk. However, we have added the following to the Discussion section: “Finally, the exposures that we did include likely have heritable and non-heritable components; exposures that are more heavily influenced by heritable factors are potentially more exposed to family-level confounding. However, the relatively small effect of unmeasured family-level confounders within the sibling participants might lessen this concern.” (pp.23-24)

5) Selection of covariates – It seems that some of the covariates selected are related to the present situation -at the time the child is 9-10 years (for example the presence or absence of a partner, screentime, family conflict, etc.). Even though these phenotypes are associated with the exposures, they could not be causing them (since they appear after the exposure in time). Rather, this adjustment could lead to over-adjustment. This is also the case when adjusting for alcohol use and smoking after pregnancy. It is likely that people smoking and drinking during pregnancy would continue to do this after pregnancy as well. By adjusting for these variables, a real effect between smoking/drinking during pregnancy would be removed. It would be interesting to see results without these adjustments as well. It would be much better to have a measure of maternal history of mental health before pregnancy, but I guess that is not available.

Thank you for this important comment, and for the opportunity to clarify/elaborate. First, to clarify, we did not include measures of smoking or alcohol use during versus after pregnancy (data are not available from ABCD to cover the period between birth and the child’s enrollment at age 9-10). We did separately examine smoking and other substance use “early” in pregnancy (defined as occurring before pregnancy was recognized) and “late” in pregnancy (defined as occurring after pregnancy was recognized.) In cases where early exposure to substances was significantly related to CBCL outcomes, late exposure was also included as a covariate, even in the minimally adjusted model, given (1) use of substances late in pregnancy was almost always preceded by their use early in pregnancy, and (2) effects of these substances late in pregnancy might differ than effects of early exposure. 

We agree that among the covariates (outside the 8 prenatal exposures of primary interest), some could be thought of as potential confounders (e.g., those related to socioeconomic status), while others might contribute directly only to the outcome measurement (CBCL) and not to the primary predictors (e.g., screen time). However, even among the latter, there are likely complex relationships that refer back to confounding variables (e.g., socioeconomic status influencing screen time). Further, as the reviewer points out, including covariates that are potential mediators could result in loss of meaningful signal.

Given the enormous potential complexity here, we decided to look at the extremes, by conducting and reporting parallel analyses – one adjusted for the bare minimum (i.e., site and late exposure to substances, as described in lines 283-289), and one fully adjusted. Critically, the similarity of results between both approaches (i.e., load effects on CBCL outcomes that are both statistically and clinically significant) suggests that the relationships between prenatal exposure load and psychopathology are robust, regardless of the number of covariates used, and of a wide range of their potential interrelated and/or confounding effects. 

We have added text to the Discussion to emphasize these points:

“We included in the full model a range of covariates, including both those that could confound relationships between prenatal exposures and CBCL scores (e.g., socioeconomic factors), and those that could not directly affect the exposure but are known to associate with CBCL (e.g., early childhood trauma). We also included variables that could contribute to psychopathology, but also potentially be affected by it (e.g., screen time). This approach also comes at the potential cost of attenuating causal effects of the adverse prenatal exposures, if, for example, some of the included covariates mediate their relationship to child psychopathology. However, the effects of exposure burden on CBCL scores were comparable regardless of the number of included covariates, as seen by comparing the minimally and fully adjusted models. This pattern suggests that the relationship between exposure load and child psychopathology is robust, regardless of the complex effects that the covariates likely confer.” (p.20)

We agree that including that maternal mental health during pregnancy would be ideal, but unfortunately these data were not collected. This is noted as a limitation in the Discussion (p.23). 

Minor comments:

Title/Abstract:

6) I would remove the word dimensional from the title (and from the objective in the abstract (line 52). As long as you state in the methods section that you use a dimensional measure of psychopathology. In addition, you also run analyses with a cut-off.

We have removed “dimensional” from the title and abstract per the reviewer’s suggestion.

7) Report betas/ORs with CIs instead of p-values. I would also recommend focusing on the results of the main aim – the cumulative effect of the exposures on dimensional outcomes (in discovery and sibling sample), and also the cumulative effect of the exposures on the binary outcome. (not on the independent effects of the various exposures)

We have replaced p-values with ORs and confidence intervals, and re-worded the results to emphasize the importance of cumulative effects:

“Results: In minimally and fully adjusted models, 5 exposures (unplanned pregnancy, maternal alcohol use early in pregnancy, tobacco use early in pregnancy, birth complications, and pregnancy complications) independently associated with significant but small increases in CBCL total score. Among these 5, none increased the odds of crossing the threshold for clinically significant symptoms by itself. However, odds of exceeding this threshold became significant with 2 exposures (OR=1.69, 95% CI 1.33-2.15), and increased linearly with each level of exposure (OR=1.33, 95% CI 1.24-1.42), up to 2.78-fold for ≥4 exposures versus none. Similar effects were observed in confirmatory analysis among siblings. Within sibling pairs, greater discordance for exposure load associated with greater CBCL total differences, suggesting that results were not confounded by unmeasured family-level effects. (pp.2-3)

Introduction:

8) Line 101 - The analogy of polygenic risk scores should be rewritten if included. Right now it is not clear to me what the message is. I guess the message is that it seems that both environmental and genetic influences are made up of the additive influence of very small influences dispersed across the genome/phenome.

Thank you for the opportunity to clarify. PRS provides a (linear) model of cumulative effects of risk factors – although because effects of individual variants are tiny, their additive effects are also small. Still, PRS provides a model to evaluative cumulative effects that is potentially useful for environmental effects – the important difference being that because effect sizes of individual exposures are so much larger than for individual genetic variants, additive effects of prenatal exposures may quickly become clinically significant. We have revised the language of this paragraph accordingly:

“Whether exposure to multiple common insults during pregnancy also exerts a dose-dependent risk for significant psychopathology remains unclear. There are few models that account for effects of multiple risk factors on such risk, although polygenic risk scoring (PRS) is perhaps the most notable. PRS studies demonstrate cumulative, linear effects across thousands of genetic variants of small effect; however, additive effects of even the strongest (e.g., genome-wide significant) common genetic variants are modest (23). In contrast, given the larger effect sizes attributed to individual prenatal environmental exposures, it is possible that only a small number of such exposures occurring in linear combination could substantially increase risk – not only for increased dimensional symptoms, but potentially for crossing the threshold into clinically significant psychopathology.” (p.5)

9) Possible mechanisms should be described in the introduction. Why would we expect a dose-response relationship?

The reviewer raises an important question, one which goes beyond our ability to answer at present – aside from suggesting that known linear effects of genetic variants provides a theoretical template for this work. Any mechanistic explanation for linear effects of environmental exposures would be speculative in these early days, which we now acknowledge explicitly in the Discussion: 

“While the present findings do not offer any specific mechanistic insights, additional translational work in this area might identify targets for early interventions that negate risk across a range of psychiatric syndromes.” (p.22) 

However, in our view the uncertainties about biological mechanisms does not diminish the potential clinical importance of the current results, as they are the first to demonstrate a strong linear association between adverse prenatal exposures and risk for psychopathology in children. 

Methods:

10) [Original] line 178 (now line 188)-Prenatal exposures – Under this heading 5 late exposures are also described. These could be removed from this description.

Late exposures were indeed extracted and coded in the same way as early exposures. Even though none of them met the 5% threshold for inclusion as a primary exposure of interest, these data were preserved in the analysis to isolate them from effects of early exposures that did meet the 5% threshold, as described later in the same section:

“Exposures present in less than 5% of the sample were dropped from the analysis, excepting one scenario: if early substance exposure occurred in at least 5% of cases, late substance exposure was also included regardless of frequency. This approach was necessary given the high co-occurrence with main predictors of interest (i.e., early use of these substances), potentially different teratogenic effects, and related confounding potential (see also Table S3). 

 Specifically, opiate or cocaine use during pregnancy (early or late) was reported in less than 5% of the sample, as were late alcohol, tobacco, or marijuana use. This resulted in using 8 adverse prenatal exposures for primary analysis (unplanned pregnancy; early alcohol, tobacco, or marijuana exposure; pregnancy complications; birth complications; preterm birth; and Caesarean section). Late exposure to alcohol, tobacco, or marijuana were treated as covariates of no interest, as above.” (p.10)

11) Why did you include only 5 imputation sets? I thought a higher number was more often used.

Per both Reviewer suggestions, we have increased to 200 imputation sets, which did not appreciably change the results. Please see response to Reviewer 2, point 13 below.

Reviewer #2: 

1) This is a study of potential interest, but as of now still contains major and moderate theoretical and methodological weaknesses. The major uncircumventable weakness is of course the retrospective report on the pregnancy. The introduction and discussion should make clear which prior studies have prospective and retrospective reporting.

We agree that prospective studies are optimal, and the retrospective report of maternal exposures is a limitation of the ABCD Study. That said, when including careful attention to potential confounding variables, retrospective studies still have value -- especially when linking prenatal events to symptoms that emerge a decade or more later, posing practical constraints on fully prospective approaches. The literature on individual exposures that is cited throughout the manuscript reflects a mixture of prospective and retrospective studies. We make both of these points more explicit in the revised Introduction:

- “Across both retrospective and prospective studies, adverse prenatal exposures that occur more frequently, such as pregnancy or birth complications (8, 9), prematurity (10), maternal infections (including both serious infections such as influenza, and more minor ones such as urinary tract infections) (11, 12), and maternal substance or tobacco use (13-17, 18-20) have also associated with a range of psychopathology, including disorders that emerge during childhood.” (p.4)

- “We leveraged baseline data from the ABCD Study to relate cumulative burden or loading of adverse prenatal exposures, obtained through retrospective report, to dimensional measures of psychopathology.” (p.6) 

Limitations posed by the use of retrospective report are extensively described in the Discussion (pp.22-23).

2) In the introduction the authors allude to common causal pathways, but for some reason study independent contributions of risk factors. This does not make sense. The authors should start out with an analysis leading to aggregation of risk factors and consider using techniques such as lasso regression and elastic net to do this. It is also paramount that the authors include a DAG of the selected covariates in the model to be clear on the hypothesized links between them internally and to the outcome. Some of the covariates (potential consequences of child psychopathology) can potentially introduce collider stratification bias. Other covariates are potential mediators; these would attenuate the estimate of the causal effect of the prenatal risk factors (like adjusting smoking for nicotine dependence when studying lung cancer).

Thank you for this comment. The primary purpose, and novel contribution, of this paper is to look at cumulative effects of a range of prenatal risk factors on a range of dimensional psychopathology symptoms. The analysis individual of risk factors was conducted as an initial step, to identify those factors that contributed independently to risk and that could therefore be studied in aggregate. 

Importantly, each of the adverse prenatal exposures that we related to CBCL scores had previously associated with increased risk for psychopathology, as specified in the Introduction. As such, we did not use lasso regression or elastic net to remove or aggregate specific factors prior to the main analysis, which examined their cumulative effects.

In regard to covariate selection, Reviewer 1 raised similar concerns. Please see response to Reviewer 1, Point 5 for a full discussion.

While we are grateful for these suggestions, we feel that a DAG that included all 30 covariates would still likely underplay the true complexity of these interactions, would go beyond the intended scope of the paper, and would not meaningfully change the primary results or their interpretation.

Introduction

3) Speculative to relate these subjects to Barker, Hunger Winter and Chinese famine. The mechanisms are very much unknown and not evaluated in this study. The potential mechanisms could be indefinite, but these are mostly highlighted because they are currently famous, not because they are the most likely model for these data. Same goes for prematurity, maternal infections.

We appreciate the reviewer’s perspective. However, we believe these citations are important not because they point to any specific mechanisms, but rather to establish that prenatal factors have been robustly associated with a variety of poor mental health outcomes. The broad audience of PLoS One may not be familiar with this work, which provides critical background and helps establish the rationale for looking at cumulative effects of prenatal exposures. We therefore elected to keep these references in place. 

4) Cite multivariate twin, family, and genomic studies on substance abuse and psychopathology.

We have added references to several such studies in the Introduction (p.6) and Discussion (p.23); see also response to Reviewer 1, Point 2).

5) The literature seems somewhat selected to support the current study, but associations are confounded with contrafactuals. Please separate the prior findings into association studies and the much smaller number of contrafactual studies (e.g. sibling designs, children of twins, IVF, and mendelian randomization). This would also bring to light the novelty of the current sibling study.

The Introduction now makes clearer the specific advantages of sibling-based studies and cites several relevant examples (p.6). Please see response to Reviewer 1, Point 2. 

6) The term pluripotent is ambiguous in this context. The authors should state more cleary what their hypotheses are and if they are also referring to non-additive/interaction effects of exposures.

We changed “pluripotent” to “non-specific” (line 119), and clarified that we were looking at effects of increased “additive” loading (line 145). 

7) Line 138: The term increased dimensional psychopathology is unclear.

We have added text as follows: “The use of dimensional measures enabled assessment of early and incompletely expressed psychopathology in school-aged children, and leveraged continuous variance in these traits across the population.” (p.6)

8) The authors state that “multiple individual adverse prenatal exposures would associate independently with increased dimensional psychopathology”. They should include in the manuscript a DAG of all these hypothesized associations. Hypotheses about causal links between the chosen covariates are of great importance when interpreting the independent contributions.

Please see response to Reviewer 2, Point 2 above. As this type of analysis has not previously been attempted, we chose a relatively simple (additive) and clinically intuitive model to assess effects of exposures that are independently associated with clinical outcomes. 

Methods

9) The term “discovery sample” does not fit well here. The sibling and non-sibling samples represent to some extent different parts of the population. Total effects should therefore also be estimated in the sibling sample.

We dropped the terms “discovery sample” and “replication sample” and replaced them with “initial sample” and “validation sample” respectively. 

10) Measures of screen time, family conflict, and some significant traumas could be consequences of the outcome (e.g. by evocative or active gene-environment correlations) leading to a potential collider stratification bias. 

Please see response to Reviewer 1, Point 5; and Reviewer 2, Point 2 above. Excluding variables (such as screen time) that are potential consequences or causes of child psychopathology leaves open the possibility that uneven distributions of these variables among exposure groups could cause spurious signal. Regardless, inclusion or exclusion of these covariates did not substantially alter the results. 

11) All covariates that are potential consequences of child psychopathology should be omitted from the analyses.

We agree that certain covariates may, in part, reflect consequences of child psychopathology – although these same factors may also contribute to it. Nonetheless, we conducted a new sensitivity analysis that only included covariates that are potential confounders (and, specifically, excluded screen time, traumatic exposure, and parental conflict (see Table S13 and S14, copied below). Excluding these factors somewhat increased effects of exposure load on CBCL outcomes (compare to Tables 2 and 3, which are reproduced below in response to Reviewer 2, Point 13).

Nonsiblings:

EXPOSURE LOAD, N Odds of CBCL total ≥60 

 Odds ratio (95% CI) p

0, N=1,650 Reference --

1, N=2,760 1.15 (0.91 to 1.45) .254

2, N=2,079 1.96 (1.55 to 2.48) <.001

3, N=986 2.72 (2.10 to 3.54) <.001

≥4, N=423 3.69 (2.68 to 5.08) <.001

Siblings:

EXPOSURE LOAD, N Odds of CBCL total ≥60 

 Odds ratio (95% CI) p

0, N=357 Reference --

1, N=479 1.71 (0.76 to 3.85) .194

2, N=356 2.40 (1.01 to 5.71) .048

3, N=146 4.71 (1.76 to 12.59) .002

≥4, N=54 6.27 (1.61 to 24.40) .008

We also added the following text to the Results:

 “In an additional sensitivity analysis, we limited covariate inclusion to those that could confound results (i.e., though direct effects on both the prenatal exposure and outcome), and as such excluded postnatal exposures (screen time, traumatic exposure, parental conflict). Results were slightly stronger than in the fully adjusted model (Tables S13, S14).” (p.18)

However, we continue to include these variables in the primary analysis, as we cannot disambiguate cause versus effect with regard to psychopathology (and in some cases, e.g., trauma exposure, there is a strong case to be made that it is mostly causal). Were these variables to differ in frequency among prenatal exposure level groups, these differences might complicate the interpretation of prenatal effects on CBCL. Regardless, again, inclusion or exclusion of these covariates does not alter the primary result of the paper, which is that multiple adverse prenatal exposures associate with linear and clinically meaningful increases in psychopathology risk.

12) Variables that are potential mediators should be only included in additional sensitivity analyses. (akin to adjusting the smoking – lung cancer association for nicotine dependence; leading to null effects).

Please refer to Reviewer 1, Point 5; and Reviewer 2, Point 2 above.

13) The authors should present more detailed information on the extent of missing data. Regardless, five imputations is most likely far from enough. Run with as many imputations as your machinery can perform (e.g. 50, 100, or 200 imputations).

Missing data are summarized in Tables S1 and S2 At the reviewer’s request, we increased the number of imputation sets to 200. 

Within the non-sibling group, as expected, given the relatively low number of missing values, few participants ended up in new exposure load groups. Accordingly, primary results were very similar using 200 imputed data sets compared to the original 5 imputed data sets:

Original manuscript (5 imputation sets):

Revised manuscript (200 imputation sets):

EXPOSURE LOAD, N Odds of CBCL total ≥60 

(minimally adjusted) Odds of CBCL total ≥60 

(fully adjusted)

 Odds ratio (95% CI) p Odds ratio (95% CI) p

0, N=1,650 Reference -- Reference --

1, N=2,760 1.27 (1.00 to 1.60) .046 1.10 (0.87 to 1.41) .411

2, N=2,079 2.24 (1.78 to 2.82) <.001 1.69 (1.33 to 2.15) <.001

3, N=986 3.31 (2.56 to 4.27) <.001 2.27 (1.73 to 2.98) <.001

≥4, N=423 4.42 (3.23 to 6.05) <.001 2.78 (1.99 to 3.89) <.001

Similarly, results persisted in the sibling group, and remained analogous to the non-sibling group:

Original manuscript (5 imputation sets):

Revised manuscript (200 imputation sets):

EXPOSURE LOAD, N Odds of CBCL total ≥60 

(minimally adjusted) Odds of CBCL total ≥60 

(fully adjusted)

 Odds ratio (95% CI) p Odds ratio (95% CI) p

0, N=357 Reference -- Reference --

1, N=479 3.34 (0.75 to 14.80) .112 1.78 (0.86 to 3.65) .118

2, N=356 4.10 (0.80 to 20.99) .090 2.48 (1.19 to 5.16) .015

3, N=146 8.53 (1.37 to 53.31) .022 3.92 (1.68 to 9.14) .002

≥4, N=54 20.71 (2.15 to 199.25) .009 3.90 (1.26 to 12.08) .018

14) The imputation analysis must be informed on the multilevel structure of the data. This is a major concern. The cites [sic] can be imputed separately (some applications have a “cluster” feature for this), and siblings can be imputed in a wide format and the flipped back to a long format.

We appreciate this suggestion and implemented the revised imputation analysis accordingly. As now described in the updated Methods: 

 “Analyses were conducted using R version 3.6.1. Missing data, assumed to be missing at random (MAR), was imputed using multiple imputation by chained equations (mice package 3.11.0), which imputes individual variables according to their own distribution and requires an imputation method to be assigned to each variable. Cluster-level effects of site were accounted for by creating separate regression coefficients for each site to be used in the imputation model (i.e. a fixed effects approach). Siblings were imputed in wide format and flipped back to long format for subsequent analyses to account for cluster effects of family. Continuous and ordinal variables were imputed with predictive mean matching and dichotomous variables with logistic regression. All variables that were included in later regressions - with the exception of transformed variables - were selected as predictors for the imputation models. 200 datasets were imputed from which parameter estimates were pooled to derive beta weights, confidence intervals, and p-values, per guidelines described by Rubin (48). This method accounts for variance both within and across the imputation sets (i.e., additional variance due to missing data). We also performed sensitivity analyses to compare group means/distributions and primary results from the full (imputed) and non-imputed data sets.” (pp.12-13) 

15) The authors should describe the type of multiple imputation used (e.g. multivariate normal).

As above, we used multiple imputations by chained equations.

16) Are there sampling weights available for the ABCD study? If so, the authors should use them.

The ABCD study was not designed to be a representative study of the US population, nor is it described that way in the manuscript. To our knowledge, sampling weights are not available. 

17) There is a large number of covariates used. The authors could consider using lasso or elastic net regression to simplify the model (e.g. available in STATA 16). Such approaches inform on the extent of out of sample replicability (such as the sibling sample).

The rationale for including adverse prenatal exposures are covariates is described above in Reviewer 1, Point 5; and Reviewer 2, Point 2. In regard to the other covariates, as described in Reviewer 1, Point 4, the sample size is sufficiently large such that there are an adequate number of subjects per variable. Further, given that the results of the minimally (1 covariate – i.e., site – in addition to the adverse prenatal exposures) and fully adjusted 15 covariates in addition to the adverse prenatal exposures) models are similar (i.e., demonstrating significant linear effects of exposure load), removal of covariates through the LASSO procedure would likely have minimal impact. Finally, use of shrinkage methods would render the effect size of results more difficult to interpret clinically. For example, we report that four or more exposures associates with an approximate three-fold increase in odds for clinically relevant symptoms, as defined using established CBCL cutoffs. While we appreciate the suggestion, for all of these reasons we believe the approach that we took is well-justified. 

18) It is not necessary to state the variance component type when having only a random intercept (site) . 21 sites is not many. The authors could also consider accounting for site effects using fixed effects (i.e. adjusting for them by dummies). This does not presume a normally distributed latent “site factor”. This opens the door to do all kinds of analyses where multi level models are not easily available (e.g. elastic net).

We included site as a random effect due to the differences in number of participants among sites, and because it provides a statistical basis for generalizing results to sites outside of the study (Feaster JD et al., Am J Drug Alcohol Abuse, 2011). The Methods text has been updated to reflect this reasoning:

“Random, rather than fixed, effect modeling was used due to variation in sample size among sites, and to provide a statistical basis for generalizing results beyond the study sites (47).” (p.12).

19) The calculation of the “integer sum of a child’s prenatal exposures” was guided by the independent contributions. This seems absurdly misguided. In the introduction the authors describe typical common pathway mechanisms for exposures. This is akin to predicting depression using the NEO-PI-R neuroticism items, but only basing the depression risk score on the items that independently predicting depression. The point here is to aggregate risk factors, not disaggregate. One blunt option is to do bivariate analyses and then calculate a risk score based on these associations. I would strongly recommend that the authors apply a sound variable selection procedure such as lasso or elastic net regression to create the risk score.

Please see responses above. We believe that additive models are justified based on the fact that each exposure has been separately considered and related to risk in previous studies, and on the fact that they are statistically independent from each other in the present sample. There are potentially numerous ways of deriving a “risk score”; we chose the one that seemed most straightforward and readily interpretable by clinicians. There are also numerous precedents for this integer approach in the adverse childhood experiences (ACE) literature, including two that we cite (Lansing et al., J Anxiety Dis 2017; Merrick MT et al., Child Abuse Negl 2017).

20) The authors use the term “clinically significant psychopathology” for >65 on CBCL syndrome scales and >60 for CBCL “broad band scales” (standard term?). According to the CBCL manual these are the threshold for “borderline clinical range”. Please revise the text accordingly. The “clinical range” is according to the manual set at a higher level.

We define “borderline clinical range” (≥93th percentile within a normative sample) as being clinically significant in this study of 9 to 10 year olds (following from numerous previous studies, e.g., Mazefsky et al., J Psychopathol Behav Assess, 2011), as now clarified: 

“Next, to assess effects of the 8 adverse prenatal exposures on odds of clinically relevant psychopathology, CBCL total scores were recoded as being within (<60) versus above the threshold for borderline clinical significance, as previously defined (37).” (p.14)

“Broad-band” is a standard term for CBCL internalizing and externalizing scores (see, e.g., Petty et al., J Anxiety Disord 22:532-539, 2007).

21) Please also include linear models with the CBCL continuous scores for comparison. This would be especially helpful in the sibling analyses, since dichotomization introduces measuring error; introducing random differences between siblings close to the threshold.

We agree and now report both continuous and thresholded analyses for both the non-sibling and sibling groups. See above, Reviewer 1, Point 2.

22) If all siblings were assessed at the same site, it is not necessary to introduce site as a random effect. They are part of the between sibling effects. You could rather just use a sandwich estimator to correct the standard errors for cite [sic] dependencies.

Per the reviewer’s suggestion, in lieu of including site as a random effect in the within-family analysis, we now use a sandwich estimator to account for heteroscedasticity induced by cluster effects of site for the within-sibling analysis (p.15). 

23) Please cluster-mean center the exposure variables and present both effects of cluster mean and the centered individual variables (keep them both in the analysis). In this way covariates at both level 1 and 2 are interpretable. The reader can then assess the familial (between) risk independently from the within family risk. (i.e. what is the risk of descending from a high-risk family).

If we understand correctly, this was already correctly applied within-family analyses. Within a given family, exposure load was recentered based on the family mean (e.g., if sib one had 1 exposure, and sib two had 4 exposures, sib 1 was given a value of -1.5 and sib 2 was given a value of 1.5. Further, mean family exposure (here, 2.5) was included as a covariate, although this between-family effect fell out of significance when including within-family in the model (lines 404-405). See also Methods, lines 318-329.

Discussion

24) The authors should in depth discuss to what extent their results are a result of retrospective reporting. What is more discuss why sibling studies using prospective designs to a lesser or no extent find an association between substance use and distress during pregnancy and child psychopathology (e.g (Eilertsen et al., 2017; Hannigan et al., 2018; Lund et al., 2019), but do find this for postnatal stress (e.g. (Gjerde et al., 2019; Lund et al., 2020). As it is now it appears that the authors have cherry-picked literature supporting their findings, when in reality the small literature using contrafactual studies goes in different directions.

We appreciate the reviewer’s concerns about presenting these studies in a more balanced way – it was not our intention to “cherry pick” or otherwise misrepresent the literature. We have included the additional studies recommended by the reviewer, in the Introduction and Discussion. While we cannot speculate the degree to which the present results are influenced by retrospective report, we have added an additional comment in the Discussion: 

“However, it is notable several prospective studies of substance exposure during pregnancy report smaller or null associations with childhood psychopathology (19, 70, 71).” (p.23)

25) Although the sibling design serves as a natural experiment for exposures during pregnancy (mothers share 100% of their own genome across pregnancies), the design does not fully adjust for child-parent effects (full siblings only share 50% of their genome). This should be made clear to the reader.

We agree and added the following text to the Discussion: “Sibling designs such as these control for systematic maternal rating bias, although they do not fully adjust for child-parent effects.” (p.24)

26) The authors should also bring to light the strengths of the sibling design in that it cancels systematic maternal rating bias.

We have implemented this suggestion; please see response to Reviewer 1, Point 2 and Reviewer 2, Point 25 above.

---

## [Decision Letter · Decision Letter 1]

9 Mar 2021

PONE-D-20-15151R1

Association of adverse prenatal exposure burden with child psychopathology in the Adolescent Brain Cognitive Development (ABCD) Study

PLOS ONE

Dear Dr. Roffman,

Thank you for submitting your manuscript to PLOS ONE. After careful consideration, we feel that it has merit but does not fully meet PLOS ONE’s publication criteria as it currently stands. Therefore, we invite you to submit a revised version of the manuscript that addresses the points raised during the review process.

We look forward to receiving your revised manuscript.

Kind regards,

Maria Christine Magnus, MPH

Academic Editor

PLOS ONE

Journal Requirements:

Additional Editor Comments (if provided):

Thank you for your revised manuscript. I agree with the reviewers that the manuscript is improved. Please address the comments from reviewer number 1.

Reviewers' comments:

Reviewer's Responses to Questions

**Comments to the Author**

1. If the authors have adequately addressed your comments raised in a previous round of review and you feel that this manuscript is now acceptable for publication, you may indicate that here to bypass the “Comments to the Author” section, enter your conflict of interest statement in the “Confidential to Editor” section, and submit your "Accept" recommendation.

Reviewer #1: All comments have been addressed

Reviewer #2: All comments have been addressed

2. Is the manuscript technically sound, and do the data support the conclusions?

Reviewer #1: Yes

Reviewer #2: Yes

3. Has the statistical analysis been performed appropriately and rigorously? 

Reviewer #1: Yes

Reviewer #2: Yes

4. Have the authors made all data underlying the findings in their manuscript fully available?

Reviewer #1: Yes

Reviewer #2: Yes

5. Is the manuscript presented in an intelligible fashion and written in standard English?

Reviewer #1: (No Response)

Reviewer #2: Yes

6. Review Comments to the Author

Reviewer #1: The authors have done a great job addressing most of my concerns. I have some additional comments/minor suggestions:

Introduction:

Page 5 (line 101) you write “risk for significant psychopathology” – the word significant should be deleted.

Page 6 (line 133) – you write “incompletely expressed psychopathology” – unclear what this means, please rewrite.

Page 6 (line 139) – “individual- and family-level variables” – delete individual?

Page 6 (line 139-141)- I would recommending rewriting the last section on page 6 to something like: “If the association is causal, siblings who are discordant for exposures should show greater differences in psychopathology. Conversely, if the discordant siblings do not differ in psychopathology, the observed association is explained by unmeasured genetic and environmental factors. The results of studies with sibling design have supported causal relationships between prenatal exposure to tobacco (40), alcohol (41), and obstetrical complications (42).”

Note: Regarding tobacco, results are conflicting, please see: https://doi.org/10.1542/peds.2016-2509

Methods:

Exposures: In line with comments from reviewer 2 I would recommend rethinking the variable selection procedure.

Covariates: I still think that covariates should be restricted to those that are related to both exposures and outcome. I suggest the new sensitivity analyses should be the main analyses (Tables S13, S14). Possible consequences and mediator effects can be investigated in separate supplementary analyses.

Discussion:

It is stated (in the response to reviewer #2) that the ABCD study was not designed to be a representative study of the US population. I am curious – what was it designed to be? To what population can the results then be generalized?

Reviewer #2: The authors have revised their manuscript well. They have followed most recommendations and given sound reasons when not. I do not have any further comments.

7. PLOS authors have the option to publish the peer review history of their article (what does this mean?). If published, this will include your full peer review and any attached files.

Reviewer #1: No

Reviewer #2: **Yes: **Eivind Ystrom

---

## [Author Response · Author response to Decision Letter 1]

23 Mar 2021

Reviewer #1

The authors have done a great job addressing most of my concerns. I have some additional comments/minor suggestions:

Thank you for your additional comments which have helped us to refine the manuscript.

1. Introduction:

Page 5 (line 101) you write “risk for significant psychopathology” – the word significant should be deleted.

Done – deleted “significant.”

2. Page 6 (line 133) – you write “incompletely expressed psychopathology” – unclear what this means, please rewrite.

Replaced “incompletely expressed” with “subdiagnostic.”

3. Page 6 (line 139) – “individual- and family-level variables” – delete individual?

Done – deleted “individual.”

4. Page 6 (line 139-141)- I would recommending rewriting the last section on page 6 to something like: “If the association is causal, siblings who are discordant for exposures should show greater differences in psychopathology. Conversely, if the discordant siblings do not differ in psychopathology, the observed association is explained by unmeasured genetic and environmental factors. The results of studies with sibling design have supported causal relationships between prenatal exposure to tobacco (40), alcohol (41), and obstetrical complications (42).”

Note: Regarding tobacco, results are conflicting, please see: https://doi.org/10.1542/peds.2016-2509

We appreciate the reviewer’s suggested edits and have incorporated them wholesale (bottom of p.5/top of p.6). In regard to tobacco, we agree that results from previous sibling studies have been inconsistent and now indicate this more clearly in the discussion (citation 51 is the Gustavson paper that the reviewer specifically mentioned): 

“Despite the smaller sample size for that analysis, this finding suggests that the relationship between prenatal exposure load and CBCL is unlikely to be confounded by unmeasured family-level differences, a possibility suggested by previous work ( e.g., (19, 51)).” (middle of p.21)

Methods:

5. Exposures: In line with comments from reviewer 2 I would recommend rethinking the variable selection procedure.

Thank you for this comment. In the previous round of review, Reviewer 2 questioned how we selected the adverse exposures that were subsequently entered into the exposure load analysis, and specifically suggested that we use lasso regression and elastic net. 

Rather, we had used multivariate regression to select exposures for additive analyses, reflecting that [1] each of the included individual exposures had been previously associated with adverse neurodevelopmental outcomes in previous studies, as well as here; [2] this approach enabled development of a clinically intuitive and straightforward risk score analysis that aggregated known risk factors while accounting for overlapping variance; and [3] given the sample size there were no concerns about overfitting and therefore no need to reduce the number of dimensions.

While there are certainly numerous potential approaches to variable selection, feel that the approach we took is appropriate and well justified, and note that Reviewer 2 found our response satisfactory in the last round of review. We therefore elected to keep this approach in the current revision, but now provide a clearer justification for this approach within the manuscript.

As now described more explicitly in the Methods (bottom of p.14/top of p.15):

“Finally, to determine the cumulative odds of CBCL total ≥60 as a function of loading for adverse prenatal exposures, the GLMM was repeated, substituting the integer sum of a child’s prenatal exposures; this loading score was calculated using only exposures that were shown to associate with increased CBCL total scores in the fully adjusted multivariate regression models (above). We used this approach to generate additive exposure loading scores because [1] each of the included individual exposures had been previously associated with adverse neurodevelopmental outcomes in previous studies (as well as in this one); [2] it enabled development of a clinically intuitive and straightforward risk score analysis that aggregated known risk factors while accounting for overlapping variance; and [3] given the sample size there were no concerns about overfitting and therefore no need to reduce the number of dimensions.”

6. Covariates: I still think that covariates should be restricted to those that are related to both exposures and outcome. I suggest the new sensitivity analyses should be the main analyses (Tables S13, S14). Possible consequences and mediator effects can be investigated in separate supplementary analyses.

We have implemented this suggestion, so the primary (“fully adjusted”) analysis now includes only covariates that are related to exposures and outcome (and, specifically, postnatal exposure to trauma, parental conflict, and screen time are now excluded). 

As a result of this change, an additional prenatal exposure, early marijuana use, became newly significant in the first-level analysis (which identified risk factors that contributed unique variance to CBCL total score). This brought the total number of such exposures to 6. Accordingly, 6-factor load score was carried forward to all subsequent analyses.

The new results are effectively unchanged. There continue to be highly significant effects of adverse prenatal exposure load on CBCL scores, measured either as a dimensional or dichotomous outcome, in the initial (non-sibling) group; these effect replicate in the validation (sibling) group; and siblings who are discordant for exposure load continue to demonstrate differences in CBCL scores, where the sibling with the larger number of exposures exhibited significantly higher CBCL total score. 

Per the reviewer’s suggestion we now include models that are additionally adjusted for postnatal exposures as sensitivity analyses (S13 and S14). 

Discussion:

7. It is stated (in the response to reviewer #2) that the ABCD study was not designed to be a representative study of the US population. I am curious – what was it designed to be? To what population can the results then be generalized?

We thank the reviewer for mentioning this point and now address it more completely. The ABCD Study was designed to be geographically diverse (21) and “generally” representative of the demographic and socioeconomic diversity of the US. However, that enrollment was based largely at academic/urban centers could introduce selection bias, and as such it is not considered “representative” of the US population in ways that other datasets (e.g., American Community Survey) might be. 

Accordingly, we added the following text to the limitations section of the Discussion:

“[N]otably, though, as the ABCD study recruited from academic centers, urban/suburban populations are better represented, and results may not generalize as well to rural communities.” (p. 25)

Reviewer #2

The authors have revised their manuscript well. They have followed most recommendations and given sound reasons when not. I do not have any further comments.

Thank you.

---

## [Editor Report · Decision Letter 2]

5 Apr 2021

Association of adverse prenatal exposure burden with child psychopathology in the Adolescent Brain Cognitive Development (ABCD) Study

PONE-D-20-15151R2

Dear Dr. Roffman,

We’re pleased to inform you that your manuscript has been judged scientifically suitable for publication and will be formally accepted for publication once it meets all outstanding technical requirements.

Kind regards,

Maria Christine Magnus, PhD

Academic Editor

PLOS ONE

---

## [Editor Report · Acceptance letter]

7 Apr 2021

PONE-D-20-15151R2 

Association of adverse prenatal exposure burden with child psychopathology in the Adolescent Brain Cognitive Development (ABCD) Study 

Dear Dr. Roffman:

I'm pleased to inform you that your manuscript has been deemed suitable for publication in PLOS ONE. Congratulations! Your manuscript is now with our production department. 

Kind regards, 

on behalf of

Dr. Maria Christine Magnus 

Academic Editor

PLOS ONE